# Local and remote moisture sources for extreme precipitation: a study of the two catastrophic 1982 Western Mediterranean episodes

Damián Insua-Costa[1], Gonzalo Miguez-Macho[1], and María Carmen Llasat[2]

[1]Non-Linear Physics Group, Universidade de Santiago de Compostela, Galicia, Spain
[2]Department of Applied Physics, Universitat de Barcelona, Barcelona, Spain

**Correspondence:** Damián Insua Costa (damian.insua@usc.es)

**Abstract.** Floods and flash floods are frequent in the South of Europe resulting from heavy rainfall events that often produce more than 200 mm in less than 24 h. Even though the meteorological conditions favorable for these situations have been widely studied, there is a lingering question that still arises: which are the sources of humidity that could explain so much precipitation? To answer this question, the regional atmospheric Weather Research and Forecasting (WRF) Model with a recently implemented moisture tagging capability has been used to analyze the main moisture sources in two catastrophic flood events occurred during the autumn of 1982 (October and November) in the Western Mediterranean area, which is regularly affected by this type of adverse weather episodes. The procedure consists in selecting a priori potential moisture source regions for the considered extreme event, and then performing simulations with the tagging technique to quantify the relative contribution of each selected source to total precipitation. For these events we study the influence of four possible potential sources: 1) evaporation in the Western Mediterranean; 2) evaporation in the Central Mediterranean; 3) evaporation in the North Atlantic; 4) advection from the tropical and subtropical Atlantic and Africa. Results show that these four moisture sources explain most of the accumulated precipitation, with the tropical and subtropical input being the most relevant in both cases. In the October event, evaporation in the Western and Central Mediterranean and in the North Atlantic also had an important contribution. In the November episode, however, tropical and subtropical moisture accounted for more than half of the total accumulated rainfall, while evaporation in the Western Mediterranean and North Atlantic played a secondary role and the contribution of the Central Mediterranean was almost negligible. Remote sources were therefore crucial: in the October event they played a similar role to local sources while in the November case they were clearly dominant. In both episodes, long distance moisture transport from the tropics and subtropics occurred mostly in mid tropospheric layers, through well-defined moisture plumes with maximum mixing ratios at medium levels.

## 1 Introduction

The Western Mediterranean Region (WMR) is characterized by a high frequency in the occurrence of torrential rainfall episodes and floods that cause severe damages, with a very high social and economic impact (Llasat et al., 2010). An analysis carried out in the framework of the HYMEX program (Drobinski et al., 2014) showed that 385 flood events (including flash-floods and urban floods) occurred between 1981 and 2010 in north-east Spain, south-east France and south-west Italy (Llasat et al., 2013).

The main mechanism generating these heavy precipitation events (HPEs) is the strong instability induced by the warm and moist air at low levels that for most of the year sits over the mild Mediterranean waters and the ensuing vigourous convection is usually triggered by the surrounding mountains or convergence lines (e.g. Buzzi et al., 1998; Rotunno and Ferretti, 2003; Llasat, 2009). Jansa et al. (2014) and Reale and Lionello (2013) showed that heavy precipitation in the Mediterranean is usually directly or indirectly related to intense, weak or moderate cyclones. Particularly, they found that in more than 80% of heavy rain cases produced in the Western Mediterranean, a cyclone was situated nearby, in a proper location for organising a warm and moist inflow into the affected area (Jansa et al., 2001; Campins et al., 2011). Most cases occur in autumn, when the combination of a still warm sea surface temperature (after a peak in late summer), and a southward displacement of the jet stream, which usually favours the appearance of Atlantic lows or cut-off lows (COLs; e.g. Nieto et al., 2005) affecting the WMR, make this season the most favourable for the development of these extreme events. For a detailed review of the most frequent atmospheric conditions resulting in Mediterranean HPEs, please refer to Dayan et al. (2015).

While factors such as strong instability or the presence of a Mediterranean low in the vicinity are commonly associated with HPEs, the concurrence of these weather features does not ensure the development of extreme precipitation. For example, in autumn, and other seasons too, the presence of Mediterranean cyclones is certainly much more frequent than the occurrence of catastrophic flooding episodes. Similarly, COLs affecting the Iberian Peninsula are more frequent in summer and located west rather than east of Iberia (Nieto et al., 2008), but heavy rainfall and floods are mainly recorded on the eastern Iberian Mediterranean shore and in autumn. Thus, an important question arises: what is the discriminating factor among many apparently similar weather situations where only one produces an HPE? The starting hypothesis of this work is that the factor setting extreme precipitation situations apart is the existence of a very large moisture supply from remote regions outside the Mediterranean. This very humid external influx, when added to local Mediterranean moisture, would yield the enormous amounts of total precipitable water (TPW) needed to produce the rain accumulations commonly recorded in these episodes, which often remind of the values associated with tropical systems. Once sufficient TPW is present, any mechanism able to concentrate and release this moisture over a small area can cause a flood-producing precipitation event. Under this hypothesis, the configuration of the large-scale circulation would therefore be also critical, since it determines whether an intense moisture transport from remote regions can be established or not.

However, in the ample literature analyzing the different contributors to the genesis of HPEs in the Western Mediterranean, moisture as a key factor is sometimes undervalued or not considered in depth, often assuming that the high values of TPW involved in these events originate locally at low levels from sea evaporation. But, where does such large amount of water vapour really come from? Is it evaporation in the Mediterranean the main source or, on the contrary, does most of the moisture in precipitation originate remotely?

There are, nevertheless, several authors that in the last two decades have used different numerical techniques to answer these fundamental questions (see Gimeno et al., 2012, for a detailed review of numerical methods used in moisture source studies). Reale et al. (2001), employing the quasi-isentropic water vapour back-trajectory method (Dirmeyer and Brubaker, 1999), showed that moisture transported by three (westward moving) Atlantic tropical systems and their extra-tropical remnants contributed significantly to the series of floods that affected the north-western and north-central Mediterranean in September

and October of 1998. Turato et al. (2004) with the same tool demonstrated that remote moisture sources, mainly the Atlantic Ocean, were crucial in the October 2000 Piedmont flood, and concluded that the contribution of evaporated moisture in the Mediterranean was lower than presumed, at around 20% of the total. Duffourg and Ducrocq (2011) studied the moisture origin and pathways in ten HPEs that took place during the autumn of years 2008 and 2009 in the French Mediterranean region. They

also used a water vapour back-trajectory technique, in this case coupled to the Meso-NH atmospheric model (i.e., on-line), concluding that when anticyclonic conditions are dominant during the 3 or 4 days prior to the HPE, the contribution of the moisture from the Mediterranean Sea is clearly dominant, whereas when cyclonic conditions prevail, remote moisture sources have a major role. Pinto et al. (2013), combining a qualitative with a backward trajectory analysis, studied a large number of events (classified in six clusters) ocurred in Northwestern Italy betwen 1938 and 2002, and found that the North Atlantic is a

relevant moisture source for precipitation, particularly important in the extraordinary cases. More recently, Krichak et al. (2015) applied a similar method for more than 50 intense cool season HPEs recorded in different parts of the Mediterranean region from 1962 to 2007. Their results highlighted the outstanding role played by tropical moisture reaching the Mediterranean from the Atlantic Ocean and the Arabian Sea. All these studies agree on the importance of the moisture contribution from remote sources, thus supporting our starting hypothesis that a very large moisture supply from regions outside the Mediterranean

is often a key factor in these types of episodes. However, practically all of these studies were carried out with Lagrangian models, based on the spatiotemporal tracking of individual fluid particles. This method, despite being very useful for its low computational cost and easy handling, presents a series of simplifications that can introduce important inaccuracies in the calculations, such as errors in particle trajectories (Stohl, 1998) or limitations in the separation between evaporation and precipitation (Stohl and James, 2004). Therefore, further work is needed in this line of research in order to obtain a more

complete knowledge about the moisture sources for these extreme rains.

     The novelty of this article is the application of a non-Lagrangian technique for the study of moisture origin in WMR HPEs. We use an online Eulerian method, generally known as water vapor tracers (WVTs) method, which is based on coupling a moisture tagging technique with a global or regional meteorological model. This tool is currently regarded as the most accurate in moisture source studies, and has only been applied to Mediterranean events by Winschall et al. (2012). These

authors analyzed the origin of moisture feeding the extreme precipitations in Piedmont in November 2002, and found that the three main sources were land evapotranspiration, evaporation from the Mediterranean and North Atlantic moisture. In the present study, we aim at applying a new WVT moisture tagging capability recently implemented into the Weather Research and Forecasting (WRF) regional meteorological model, the so-called WRF-WVT tool. This implementation has been thoroughly validated (Insua-Costa and Miguez-Macho, 2018), showing that the method presents a high accuracy, and thus it will allow us

to quantify the contribution of different moisture sources and to perform a detailed three dimensional separation of water vapor from different origins in the development of HPEs in the Mediterranean.

     Precisely, we will apply the method to two infamous HPEs occurred in the NWMR (North Western Mediterranean Region) during the autumn of 1982. The selection of these two cases is mainly based on the enourmous socioeconomic impact they had, which is why even today they are well remembered by the population. Both events appear, for example, in the list of major flood

disasters in Europe between 1950 and 2005 (Barredo, 2007) and are still present in the scientific community and the media.

The first of these episodes occurred in October and particularly affected the Spanish Levant area. The highest precipitation amounts were observed on days 19, 20 and 21, especially on day 20, with a maximum of 426 mm fallen in Cofrentes (Valencia, Spain). Particularly dramatic was the situation in the vicinity of the Tous dam, since the exceptionally intense precipitation recorded in the river Júcar basin (where the dam is situated) caused its rupture, seriously aggravating flooding downstream. The consequences were catastrophic; there were 40 fatal victims and about 630M\$ (uninflated) in economic-losses (Barredo, 2007). The second event took place only a few days later, between November 6 and 8, with special intensity on the 7th. In this occasion, precipitation affected especially the northeast of Spain (Catalonia), Andorra and the south-east of France, with remarkable amounts such as the 408 mm recorded in Valcebollère (French Pyrenees) and 342 mm in La Molina (Catalan Pyrenees), both in 24 hours. The consequences of the event were also catastrophic; 42 casualties, adding the victims of Spain, Andorra and France (Trapero et al., 2013), and about 300M\$ (uninflated) in damages in Catalonia alone (Llasat et al., 2013). A notable feature of these two episodes is that they represent the two most common atmospheric circulation patterns associated with HPEs in the NWMR (see AP3 and AP13 weather types in the classification of Romero et al., 1999), so the conclusions obtained in this work could be extrapolated to many other cases.

The study is structured as follows: Section 2 describes the methodology and the data used, where in addition it will be presented in a more detailed way the WVT method and the WRF-WVT tool. Section 3 and 4 show the results obtained by applying the method to the cases of October and November 1982, respectively, and finally, Section 5 contains a summary and conclusions of the work.

## 2   Methods

### 2.1   The WVT method and the WRF-WVT tool

From a physical point of view, the WVT method can be conceptualized as the release of a dye within the hydrological cycle representation of a meteorological model. Moisture originating from a particular source is traced until it leaves the simulation domain or precipitates, thus making it possible to know in detail the contribution of the considered source to total precipitation at any point in a given model grid (Fig. 1).

From a mathematical point of view, the WVT method consists in replicating for moisture tracers the prognostic equations for total moisture. The equations for tracers are thus in Eulerian form, fully coupled to the full moisture equations, and must be solved simultaneously with them, i.e. "online". The reason for the latter is that in tracer calculations, eddy diffusivities in turbulent mixing are the same as those for full moisture, and in convection and microphysics processes, phase changes among the different tracer species occur as for their full moisture counterparts, but in amounts proportional to the tracer fraction in the species undergoing the change. The WVT method is therefore an online Eulerian moisture tracking strategy, highly accurate and distinct from the most commonly used Lagrangian particle tracking methods, which are integrated offline. For specific details of the implementation of the WVT method in WRF that we use here (WRF-WVTs) and its validation, please refer to Insua-Costa and Miguez-Macho (2018).

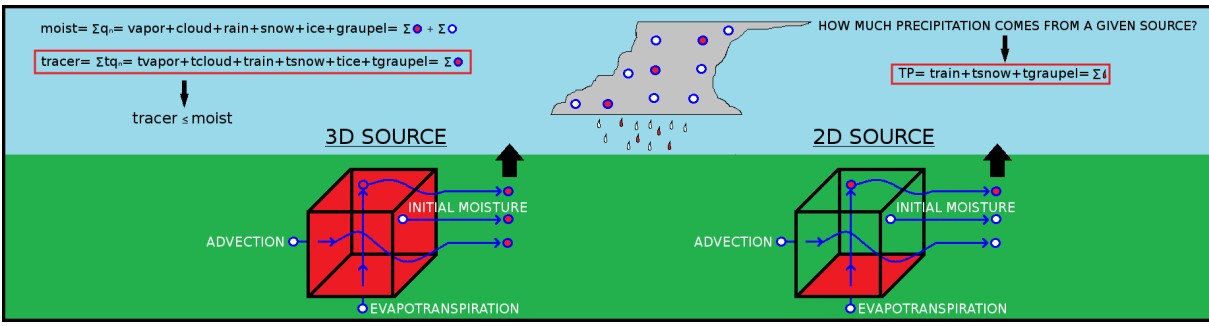

**Figure 1.** Sketch representing the fundamentals of the moisture tracer method, including the tagging of 3D and 2D moisture sources (from Insua-Costa and Miguez-Macho, 2018).

Among the different scheme options available in WRF, moisture tracking is currently implemented in the Yonsei University (YSU; Hong et al., 2006) PBL scheme, the WRF-Single-Moment 6-class (WSM6; Hong and Lim, 2006) microphysics scheme and the Kain-Fritsch (Kain, 2004) convective parameterization. Therefore, it is mandatory to choose these three parameterizations when working with WRF-WVTs, although in a convective-resolving scale, tracers can also be used without the Kain-Fritsch parameterization. In accordance with these parameterization choices, six tracer species are considered, namely tracer water vapor, cloud water, rain, snow, ice and graupel. In addition, there are also four new variables corresponding to the different types of tracer precipitation (tracer convective rainfall, tracer stratiform or grid-resolved rainfall, tracer snowfall and tracer graupel).

WRF-WVTs allows moisture tracking from two-dimensional (2D) and three-dimensional (3D) sources (Fig. 1). A 2D source refers to tagging moisture from surface evapotranspiration over a certain area. For its part, a 3D source encompasses the entire atmosphere over a region of interest, or only a part of it (for example the stratosphere), from which all exiting moisture is tagged.

## 2.2 Experimental design

We consider four source regions, three two-dimensional and one three-dimensional. The three 2D source regions cover the Western Mediterranean, the Central Mediterranean and the North Atlantic evaporative sources respectively, whereas the 3D source region tags moisture advected from the tropical and subtropical Atlantic and from tropical Africa (Fig. 2a). The 2D sources target sea evaporation; however, the tropical and subtropical regions are taken as a 3D source in order to include both evaporation and atmospheric water transport from further possibly relevant tropical or subtropical areas outside the model grid, such as the Gulf of Mexico, which is a relevant moisture source for precipitation in the WMR according to different climatic studies (Gimeno et al., 2009; Nieto et al., 2010). Special care has been taken not to tag humidity from any source twice. For example, moisture evaporated in the North Atlantic is only considered once, even when it reaches the Iberian Peninsula after traversing the 3D subtropical source region. Finally, we note that we do not contemplate all possible moisture sources, such as

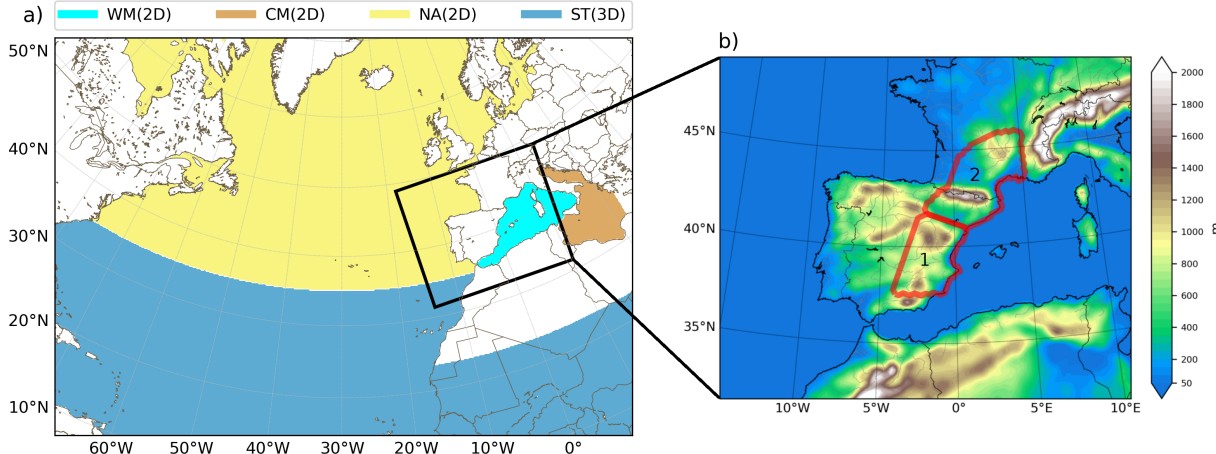

**Figure 2.** (a) Simulation domain and moisture sources considered: Western Mediterranean (light blue), Central Mediterranean (brown) and North Atlantic (yellow) two-dimensional sources and tropical and subtropical three-dimensional source (dark blue). (b) Domain for precipitation analysis with topography (m) in shades. The areas highlighted in red are the most affected by the October (1) and the November event (2).

land evapotranspiration from different continental regions. We assume that in autumn it is very diminished and hence it does not have a potentially important contribution (e.g. Sodemann and Zubler, 2010; Drumond et al., 2011).

With this sources' selection, we will be able to clarify the origin of moisture on the large scale only. In other words, we can determine whether moisture is of local or remote origin, but we will not be able to ensure, for example, where exactly in the Atlantic or tropics this humidty mostly comes from. We could subdivide the four selected sources into many more and then achieve much more detail, but for each selected moisture source a separate simulation must be carried out, with the corresponding increse in comutational cost. For example, for $1 \times 1$ degree source regions, this means hundreds of simulations just for one case. The selection proposed here is based on the choice of quite extensive sources, which does not mean they are not enlightening: a distinction is made between local (Mediterranean) and remote (Atlantic) humidity; within the remote we distinguish between tropical and non-tropical and within the local between Western and Central Mediterranean.

Simulations for both events start 10 days before their respective main date (October 20 and November 7), thereby allowing moisture sufficient time to evaporate and travel to the area affected by extreme rainfall (highlighted in red in Fig. 2b). Furthermore, this 10-day period roughly coincides with the average residence time of water vapor in the atmosphere (e.g. Trenberth, 1998; Van Der Ent and Tuinenburg, 2017); thus we can neglect the contribution of the moisture present at initial time in the atmospheric volume of the considered domain. The total time span of the experiments is 12 days.

### 2.3 Model configuration and data used

The simulations for the two 1982 HPEs are performed with the WRF model version 3.8.1 (Skamarock et al., 2008) using a single domain of 20km horizontal resolution and 35 vertical levels. Initial and boundary conditions were obtained from ERA-

Interim reanalysis (Dee et al., 2011) with spatial resolution of 0.7º and updated every six hours. In addition to the YSU boundary layer parameterization, WSM6 microphysics scheme and Kain-Fritsch convective parameterization (required when the WRF-WVT tool in its current version is activated), we have also used the Rapid Radiative Transfer Model (RRTM; Mlawer et al., 1997) and Dudhia (Dudhia, 1989) schemes for long and shortwave radiation, respectively, and the Noah Land Surface Model (Noah LSM; Chen and Dudhia, 2001). Spectral nudging of the synoptic circulation in the grid (about 1000km wavelength and longer) towards reanalysis has been applied to avoid distortions due to the interaction between the model's solution and the lateral boundary conditions (Miguez-Macho et al., 2004). Moisture and tracer advection are calculated with the 5th order Weighted Essentially Non-Oscillatory (WENO; Liu, 1994) scheme with positive definite limiter. Finally, for model rainfall validation we use the MESCAN (from Mesoscale Analysis; Soci et al., 2016) system, which combines a downscaled reanalysis and interpolated rain gauge mesurements to get a high resolution (5.5 km) daily precipitation dataset. This product is recently available in the ECMWF MARS (Meteorological Archival and Retrieval System) and covers our entire area of study.

## 3   The October event

### 3.1   Synoptic situation and precipitation

The October 1982 case, also known as the Tous event, was associated with a cold-core COL, which had originated from an Atlantic trough and was centered aloft over Morocco on the 20th, the main day of the episode (Fig. 3b). This configuration caused a marked increase in instability and the emergence of dynamic forcings favouring the appearance of upward air motions in the Spanish Levant area, the one most affected by the torrential rains. At lower levels, the cyclone consisted of an extensive low-pressure system with center over Algeria, which organized a relatively warm (Fig. 3a) and very humid (Fig. 3b) easterly flow almost perpendicular to the coast, increasing the chances of heavy precipitation. In Fig. 3b, the high amount of TPW on the east coast of Spain is particularly noteworthy, with values well above 30 mm. All these elements provided a quasi-ideal scenario for the occurrence of deep moist convection. In fact, during October 20, a mesoscale convective complex (Maddox, 1980), the first identified in Europe, developed east-southeast of the Iberian Peninsula, ultimately causing the HPE (although it was finally defined as a mesoscale convective system, MCS, due to its minor dimensions, Rivera and Riosalido, 1986). For a more in-depth analysis of the factors contributing to this event, please refer to Romero et al. (2000).

Figure 4 shows the observational analysis (Fig. 4a) and simulated (Fig. 4b) precipitation during the days of the event (October 19, 20 and 21). As mentioned earlier, the most affected region by the HPE was the Spanish Levant area and especially the Valencian Community, with maximum precipitation accumulations above 250 mm towards the interior of this region. Note that the recorded amounts in some stations were actually much higher; however, localized peak values are smoothed out in the analyzed precipitation field, since it has a resolution of 5.5 km. Precipitation was well organized around this maximum, which is consistent with the fact that the rains were produced by an almost stationary MCS. The simulated precipitation shows a very good agreement with the observational analysis, both in amounts and spatial distribution. Therefore, despite some discrepancies, we conclude that the model reproduces the episode realistically.

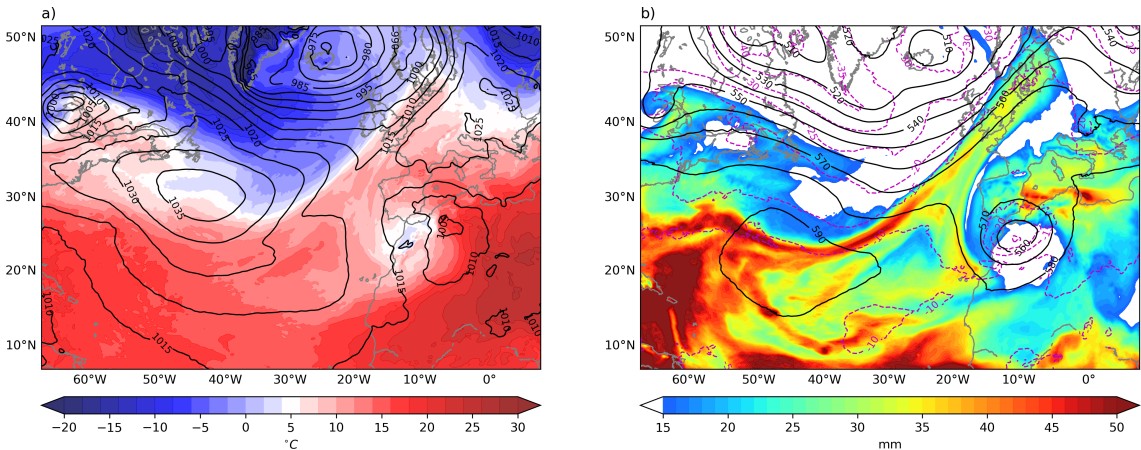

**Figure 3.** Synoptic situation (from WRF simulation) on October 20, 1982, at 12:00 UTC. (a) Mean sea level pressure (contours, hPa) and 850 hPa temperature (shades, ∘C). (b) Geopotential height (solid black contours, dam) and temperature (magenta dashed contours, ∘C) at 500 hPa and total precipitable water (shades, mm).

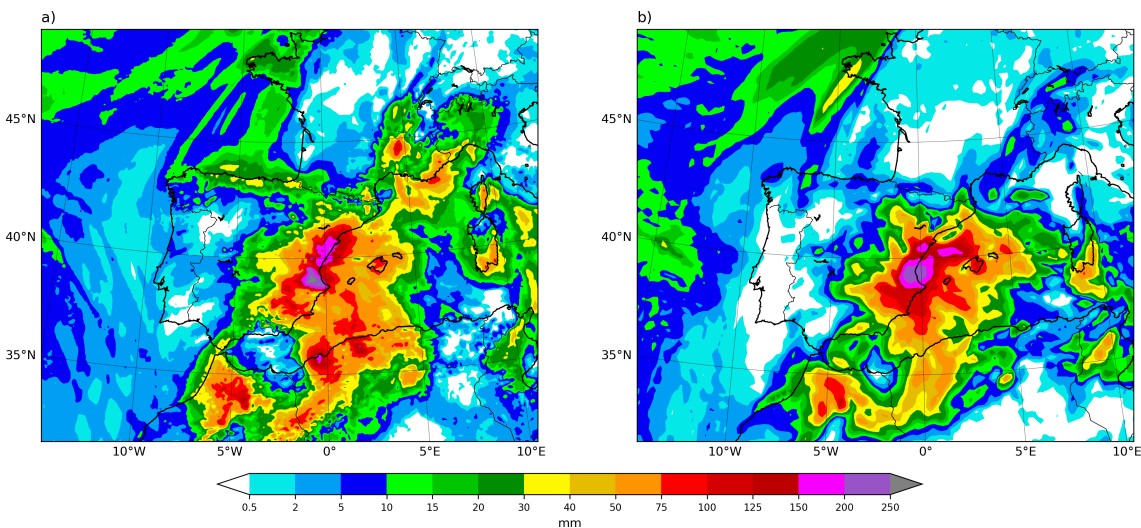

**Figure 4.** (a) Observed (from MESCAN analysis) and (b) simulated total precipitation (mm) from October 19 at 06:00 UTC to October 22 at 06:00 UTC.

## 3.2 Moisture origin

Figure 5 shows at 12:00 UTC on October 20, the TPW originating from the different moisture sources considered during the previous 10.5 days, i.e. from the beginning of the simulation (00:00 UTC, October 10). Moisture from evaporation in the Western (Fig. 5a) and Central Mediterranean (Fig. 5b), with total content values in the 5-10 mm range in both cases, remains stagnant in the Mediterranean area, suggesting that throughout the period before the event, the flow was weak in the region as a

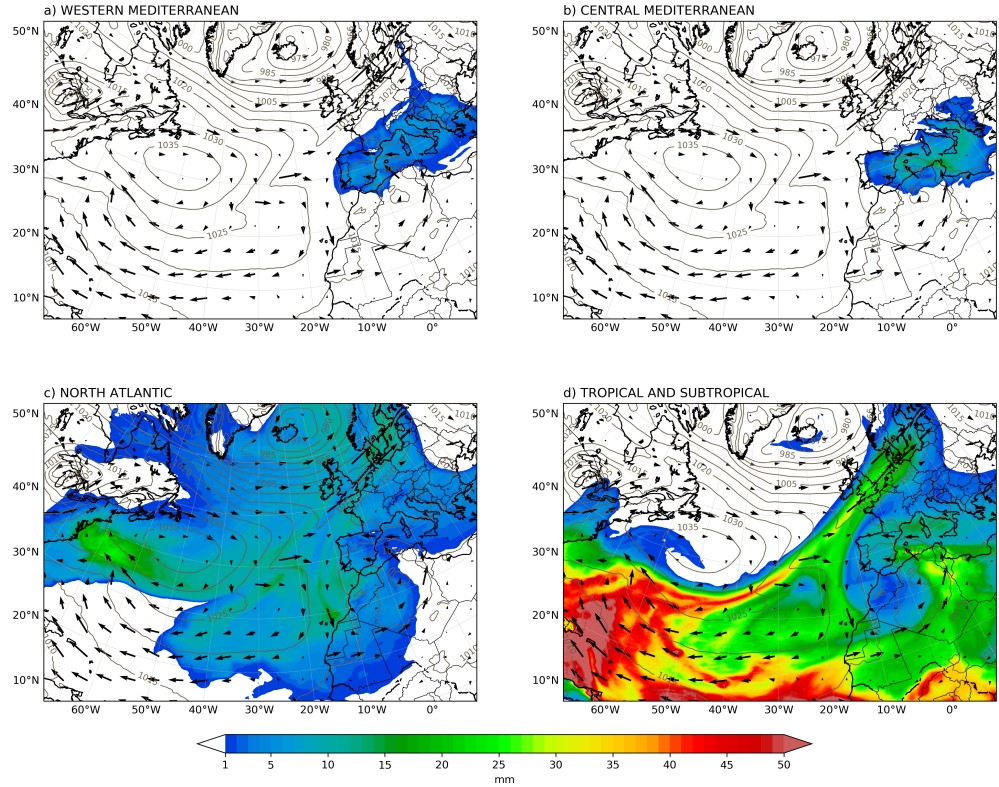

**Figure 5.** Total precipitable water (mm) coming from the Western Mediterranean (a), the Central Mediterranean (b), the North Atlantic (c) and from the tropical and subtropical Atlantic along with tropical Africa, on October 20 at 12:00 UTC. Contours show mean sea level pressure (hPa) and arrows show the vertically integrated moisture flux (kg m$^{-1}$ s$^{-1}$).

result of the prevailing anticyclonic situation. The low pressure system situated over North Africa blocks the direct advance of evaporated moisture from the North Atlantic toward the Spanish Levant area (Fig. 5c). Notwithstanding, some of this humidity reaches the region by making its way around the cyclone, and the attained values of TPW from this source are still significant, of around 5 mm. The most important contribution from any source corresponds, however, to that of moisture advected from

5 the tropics and subtropics (Fig. 5d). Following the circulation around the low in North Africa, a well-defined moisture plume rising across the Sahara reaches the east coast of Spain, yielding values of TPW of around 15 mm; locally even exceeding 25 mm.

Figure 6 depicts the source-separated vertical distribution of water vapour 12h before (00:00 UTC, October 20) and 12h after (00:00 UTC, October 21) the time in Fig. 5. Both absolute and relative contribution from each source are reflected. The

10 values shown are spatial averages over the area most affected by the event, highlighted in red and labelled as 1 in Fig. 2b. At the early stages of the episode (Fig. 6a and  6c), the atmospheric moisture content is dominated by evaporative input from the Western Mediterranean and the North Atlantic, and by advection from the tropics and subtropics, with the role played by moisture from the Central Mediterranean being negligible. At the lowest levels of the atmosphere, evaporation from the

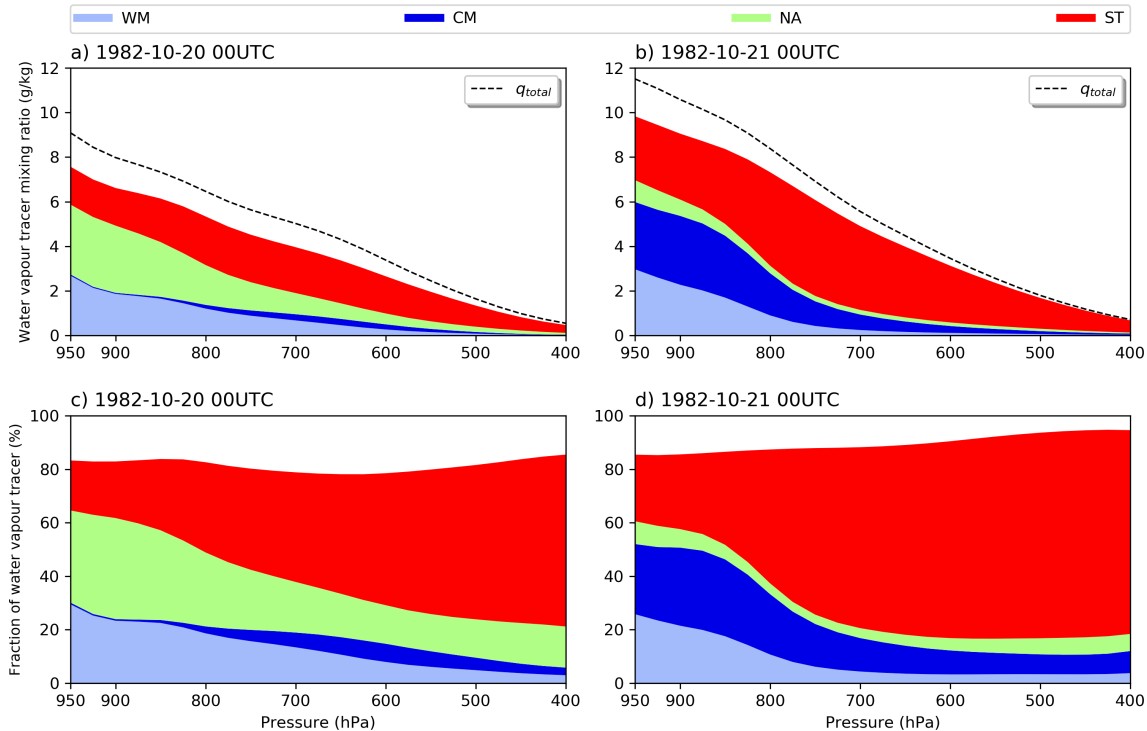

**Figure 6.** Vertical distribution of water vapour coming from the Western Mediterranean (light blue), the Central Mediterranean (dark blue), the North Atlantic (light green) and from the tropical and subtropical Atlantic along with tropical Africa (red). First-row shows absolute values (g/kg) on October 20 (a) and 21 (b) at 00:00 UTC. Second row depicts relative values (%) on October 20 (c) and 21 (d) at 00:00 UTC. Black dashed lines indicate the total water vapour mixing ratio, from considered and not considered sources (g/kg). Values are area averages over the region highlighted in red and labelled 1 in Fig. 2b.

Western Mediterranean and the North Atlantic in conjunction represent more than 60% of the existing total water vapour. Above 800 hPa, however, moisture becomes increasingly of tropical and subtropical origin, and above 500 hPa these remote sources account for more than 50% of total humidity. As the dynamics of the event progresses, one day later (Fig. 6b and 6d) the vertical distribution of moisture source contribution changes substantially. With the settling in of easterly flow induced by

5     the wide low pressure system over North Africa, moisture content from the North Atlantic becomes almost negligible and it's replaced by Central Mediterranean evaporation. In addition, the injection of tropical and subtropical water vapor is reinforced, clearly becoming the most relevant source in this phase of the event; its presence is very significant in the entire atmospheric column, accounting for more than 60% of the total moisture above 800 hPa. At this stage, the large amount of water present in the atmosphere at all levels is striking, with a mixing ratio of about 12 g/kg at 950 hPa. Finally, we note that the relative

10     combined contribution of the four sources considered is always higher than 80% throughout the entire column, which agrees with our original hypothesis that other possible moisture sources are of minor importance.

## 3.3 Precipitation origin

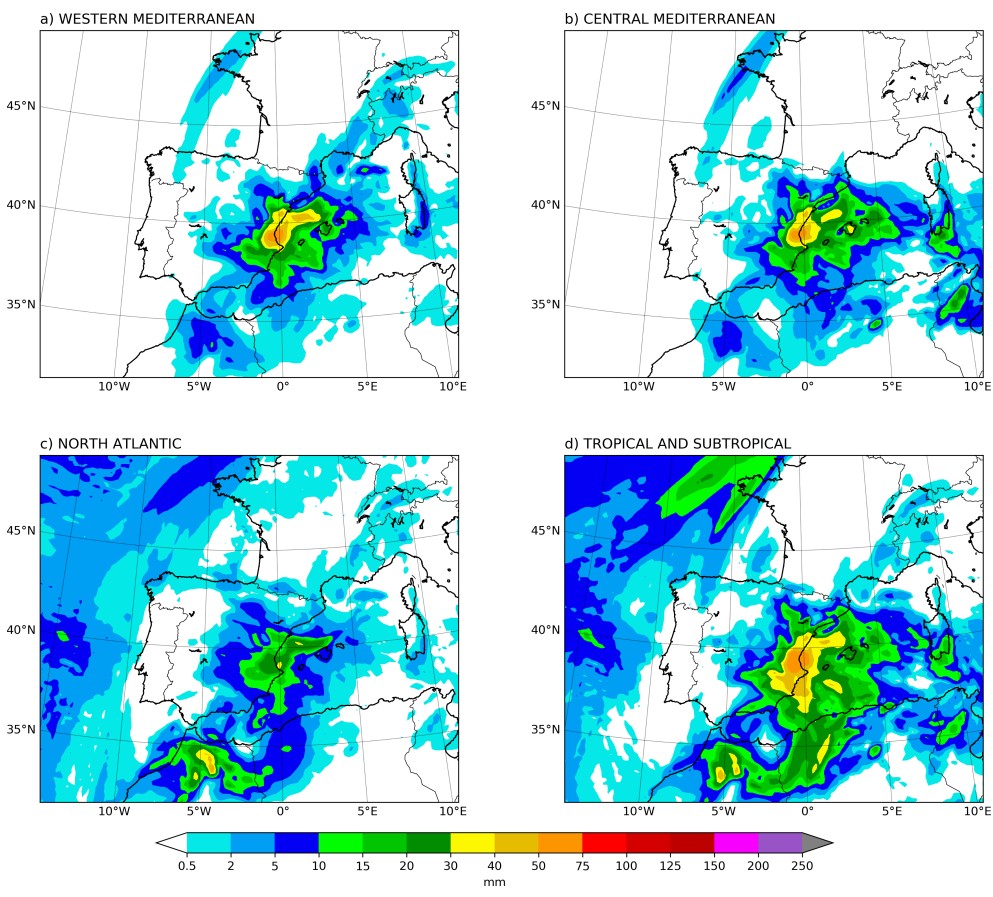

**Figure 7.** Simulated precipitation (mm) coming from the Western Mediterranean (a), the Central Mediterranean (b), the North Atlantic (c) and the Tropics and Subtropics (d) from October 19 of at 06:00 UTC to October 22 at 06:00 UTC.

From the previous analysis, it is apparent that moisture at low levels is dominated by evaporative sources, either local (Western Mediterranean) or more distant (first from the North Atlantic, later from the Central Mediterranean), while in mid and upper layers it is mostly of remote tropical and subtropical origin, more so as the event develops. Furthermore, the contribution of this advected moisture from lower latitudes increases significantly the water vapor content throughout the column. We examine next how TPW from each origin translates into precipitation, to address the main question that we posed in this study: how much of the accumulated rainfall in the event is coming from the different analyzed sources. Figure 7 shows a decomposition of the total precipitation field in Fig. 4b according to moisture origin. The contribution from the Western (Fig. 7a) and Central (Fig. 7b) Mediterranean is approximately equal, with maximum accumulations from October, 19 to 21 exceeding 50 mm in the Spanish Levant area. Here, the amounts coming from North Atlantic evaporation (Fig. 7c), albeit significant, barely reach 30 mm. In North Morocco, another of the impacted regions, the contribution of this source is, however,

somewhat higher. Rainfall from tropical and subtropical origin (Fig. 7d) represents the largest share of the total in virtually the entire area affected by the event, with values well above 50 mm over a wide swath around the location of maximum precipitation in Spain.

The relative contribution of the different sources to total precipitation during the main days of the event are quantified in Table 1. Values are calculated over the Spanish Levant area -outlined in red and labelled 1 in Fig. 2b- and shown as percentage of total rainfall. Local moisture from evaporation in the Western Mediterranean basin accounts for only about 20% of precipitation. If we expand the concept of "local" to include the Central Mediterranean, then the contribution from local sources practically doubles, to represent around 40% of the total. In contrast, at least 46% of precipitation originates from water evaporated in remote regions, with tropical and subtropical moisture being the most relevant (31% of the total). The four considered sources account for most of the collected rainfall, around 83%, consistently with the values seen in the previous section for water vapour throughout the atmospheric column.

**Table 1.** Relative contribution (%) of the considered moisture sources to the accumulated precipitation from October 19 at 06:00 UTC to October 21 at 06:00 UTC in the most affected area (region 1 in Fig. 2b).

|  | Western Mediterranean | Central Mediterranean | North Atlantic | Tropical and Subtropical |
|---|---|---|---|---|
| **Relative Contribution (%)** | 19,14 | 18,28 | 14,89 | 31,02 |

## 4  The November event

### 4.1  Synoptic situation and precipitation

As the October episode, the case of November had a very high social and economic impact, but the weather conditions leading to it were very different. There was neither COL nor cold air aloft in the most affected regions by extreme precipitation (northeast Spain and southeast France); instead, the HPE was connected with a strong omega block pattern (Fig. 8b). At 12 UTC November 7, the main day of the event, an extensive upper-level ridge associated with a strong surface anticyclone covered a large part of Europe, while a deep trough was located west of the Iberian Peninsula, thus leaving northeastern Spain and southwestern France in the frontal zone on its leading side. At the surface (Fig. 8a), a very deep low-pressure system located off the coast of Galicia organized a very intense, persistent (due to the block pattern) and relatively warm low-level south-southwesterly flow into the most affected regions. Another crucial feature drawing attention in Fig. 8b is the very high values of TPW in much of the eastern half of the Iberian Peninsula, seemingly transported to the region by an atmospheric river, which favoured the high accumulations of rainfall. All these elements indicate that dynamic rather than thermal factors were the most relevant in this case. For a more in-depth analysis of the development of this event, please refer to Trapero et al. (2013).

Figure 9 shows the observational analysis (Fig. 9a) and simulated (Fig. 9b) precipitation during the main days of the event (November 6, 7 and 8). The spatial pattern in Fig. 9a indicates that orography played a very important role, since the maximum

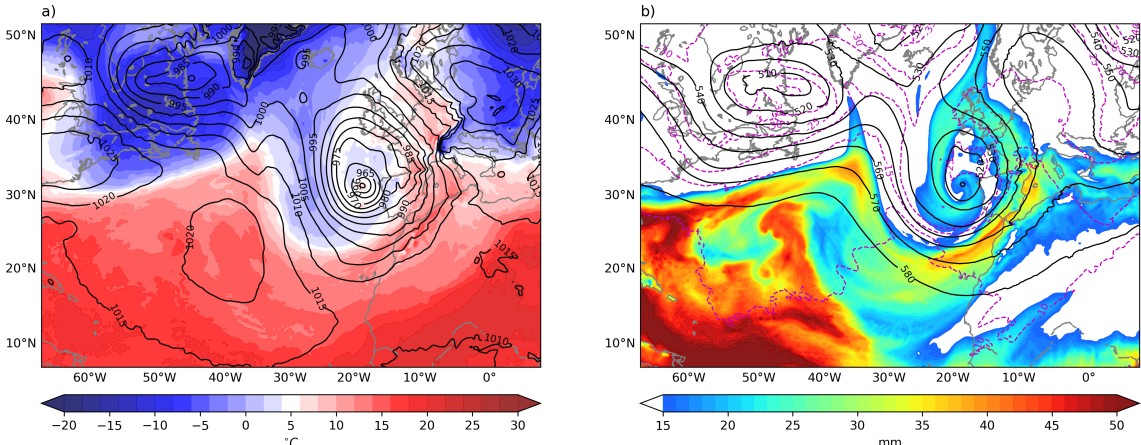

**Figure 8.** Similar to Fig. 3 but for November 7, 1982, at 12:00 UTC.

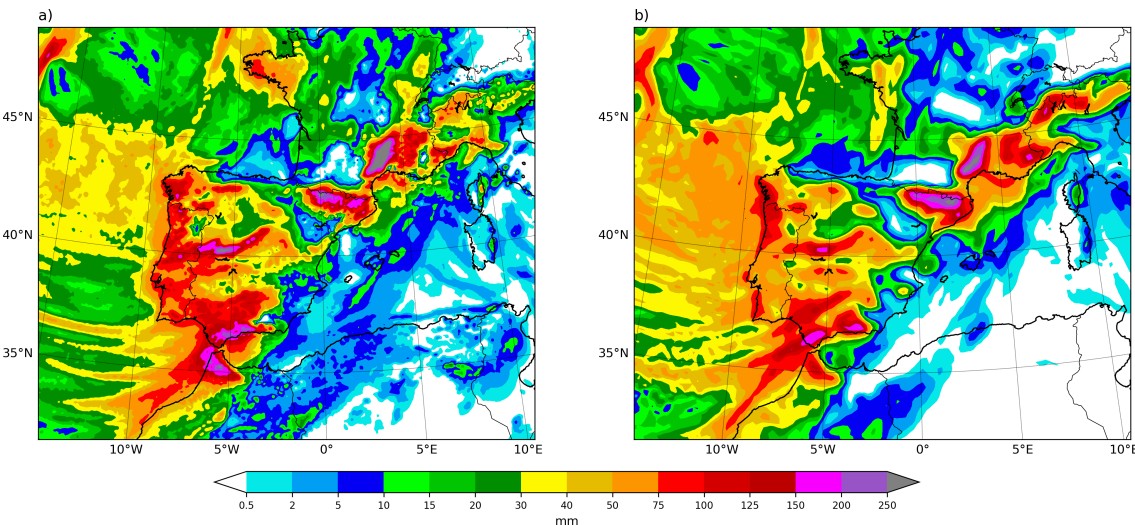

**Figure 9.** Similar to Fig. 4 but from November 6 at 06:00 UTC to November 9 at 06:00 UTC.

precipitation occurs in mountainous areas. This is especially evident in the Pyrenees and the southern section of the French Massif Central, where the highest rainfall accumulations were recorded. Precipitation peaks in the latter mountain ranges are well above 250 mm, although, as in the October case, there were much higher amounts measured at specific locations (exceeding 400 mm in just 24 h) that are smoothed out in the analysis. In this November event, extreme precipitation affected, nevertheless, a very large region, including the Iberian Peninsula, Morocco and Southern France, and was not so local as in the episode from the previous month. This suggests that the nature of precipitation was very different in both cases; in October, it was associated with deep convection whereas in November, precipitation was mainly stratiform, with strong embedded convective cells triggered by the terrain in mountain areas. Therefore, the persistence (forced by the block pattern) and orographic

lift enhancement of precipitation, together with a good supply of moisture, were the key factors in this episode. The model simulates realistically these processes and captures the actual spatial distribution and total accumulations of rainfall closely (Fig. 9b).

## 4.2  Moisture origin

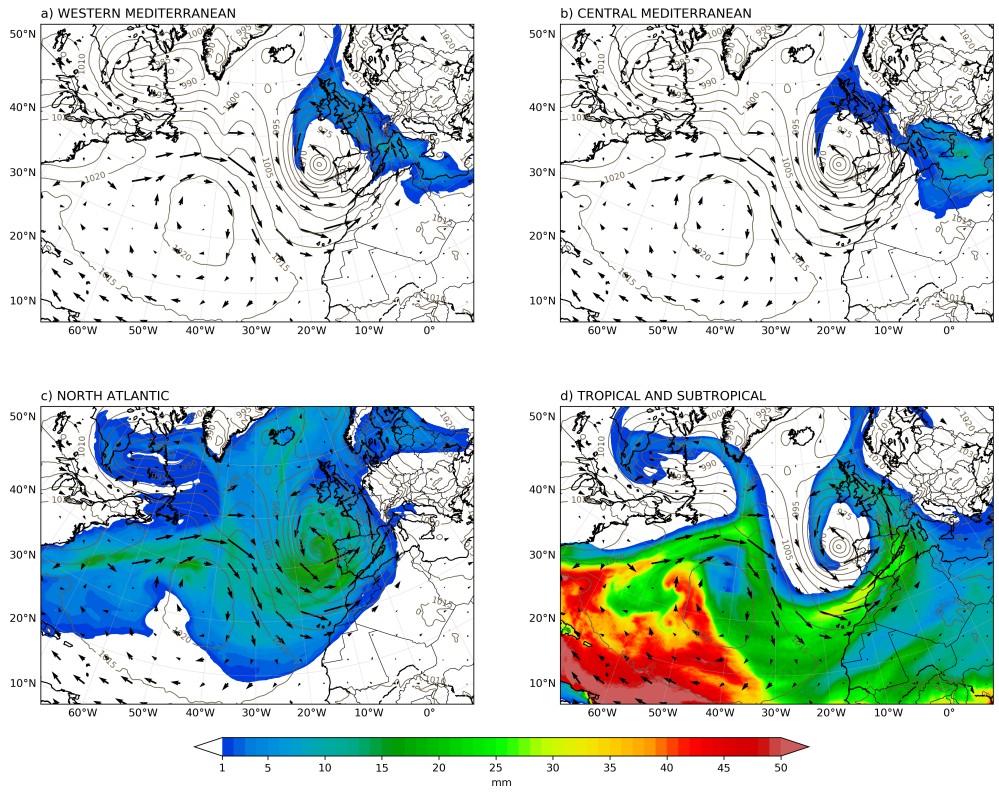

**Figure 10.** Similar to Fig. 5 but for November 7 at 12:00 UTC.

5    Figure 10 shows at 12:00 UTC, November 7, the TPW generated from each considered origin from the beginning of the simulation, 10.5 days before (October 28, 00:00 UTC). The deep low-pressure system located off the coast of Galicia picks up moisture from all the sources and redistributes it in different ways. TPW from evaporation in the Western (Fig. 10a) and Central Mediterranean (Fig. 10b) is advected due northwest, across France and the British Isles and finally transported into the Atlantic following the cyclonic circulation around the low. The Iberian Peninsula lies only marginally within this path, and as a result,

10   the amount of TPW from the Western Mediterranean is small there, less than 5 mm in Catalonia, and negligible for moisture from the Central Mediterranean. However, in southeast France, the other region most affected by the rains, the contributions from these two sources are substantially more relevant, with values of more than 10 mm of western Mediterranean TPW in the vicinity of the Gulf of Lion. Meanwhile, North Atlantic moisture is transported in large amounts toward the Iberian Peninsula

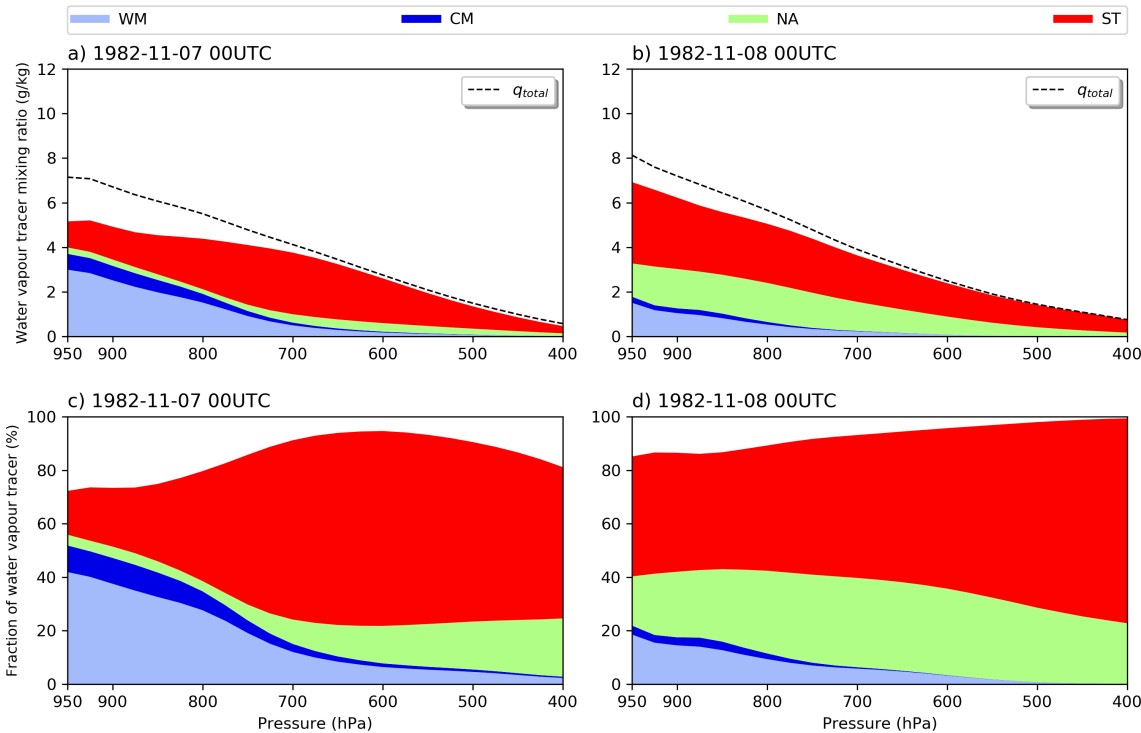

**Figure 11.** Similar to Fig. 6 but on November 7 (a, c) and 8 (b, d) at 00:00 UTC. The analysis is now over region 2 in Fig. 2b.

by the intense south-westerly flow associated with the low (Fig. 10c), and TPW from this origin attains values of around 15 mm in the western Iberian margin. Some of this Atlantic water vapor extends to the Mediterranean and France with diminished amounts of TPW, below 10 mm. Finally, as in the October case, the most important contribution to TPW corresponds to that of moisture advected from the tropics and subtropics (Fig. 10d). A well-defined moisture plume or atmospheric river enters the

5 Mediterranean through the Strait of Gibraltar, stretches along the east coast of Spain and reaches the south of France, leaving values well in excess of 20 mm of TPW in some of these areas.

The vertical distribution of water vapour from the different sources is shown in Fig. 11, which is analogous to Fig. 6 for the October case. The analysis is now performed over the region labelled 2 in Fig. 2b, the one most affected by the torrential rains. At the beginning of the episode (November 7 at 00:00 UTC, Fig. 11a and 11c) , there is mainly moisture from only

10 two origins: Western Mediterranean evaporation, dominating at low layers below 800 hPa, and advected water vapor from the tropics and subtropics, becoming predominant in mid and upper layers above that level. At a more advanced stage of the event, on November 8 at 00:00 UTC (Fig. 11b and 11d), Western Mediterranean evaporation remains in the boundary layer and loses importance while North Atlantic water vapor gains relevance throughout the column. For its part, tropical and subtropical advection becomes clearly the most abundant type of moisture at all levels. At this late stage of the event, these three sources

alone account for about 90% of TPW. Central Mediterranean evaporation and other sources not considered are irrelevant. The important contribution of remote moisture transport from the Atlantic (including the tropics and subtropics) at mid and upper

levels corroborates the hypothesis made from qualitative observations in the first in depth investigation of this event (Llasat, 1987, 1991). Finally, we note that mixing ratios are high throughout the entire atmospheric column, reaching 8 g/kg at 950 hPa; a significantly lower value, nonetheless, than in the October case.

## 4.3 Precipitation origin

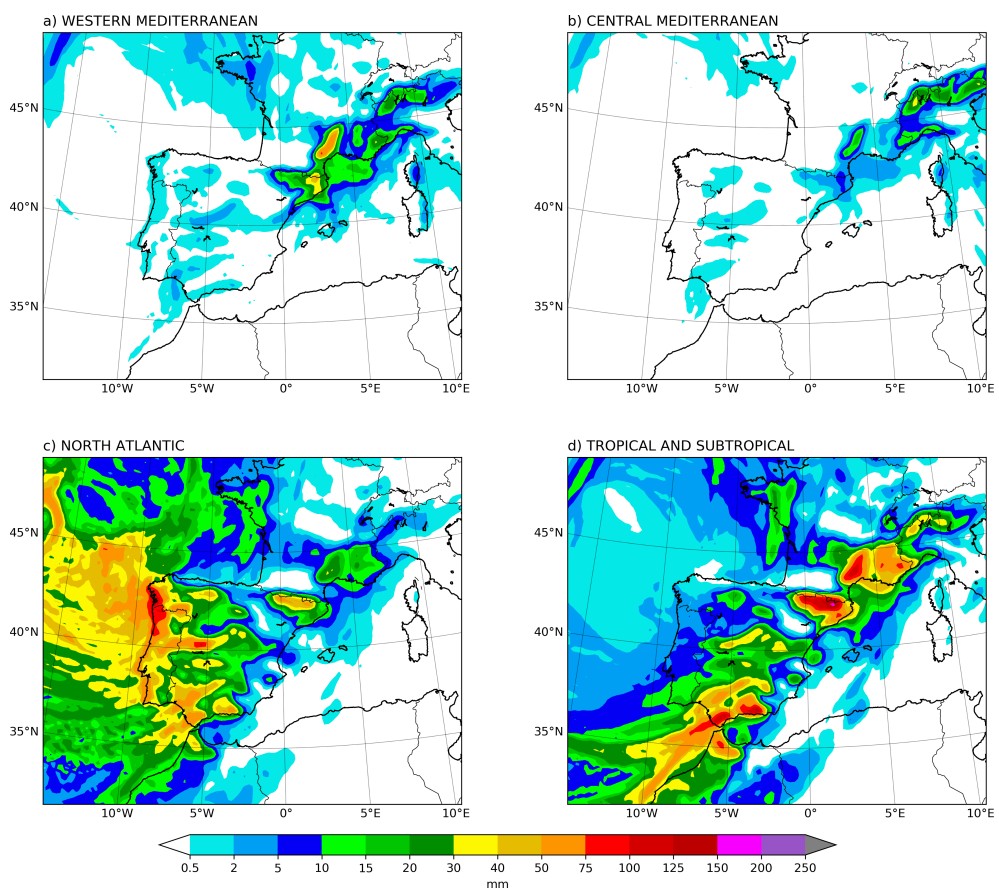

**Figure 12.** Similar to Fig. 7 but from 06th of November at 06:00 UTC to 09th of November at 06:00 UTC.

With regards to the origin of precipitation, Figure 12 shows the share corresponding to each considered source. The largest contributions are clearly from North Atlantic (Fig. 12c) and tropical and subtropical moisture (Fig. 12d). North Atlantic water vapor is found in significant amounts in rainfall in all the affected areas, and it's by far the dominant source in the western half of the Iberian Peninsula, the most exposed to the west-southwesterly flow of the storm off shore. Precipitation of tropical and subtropical origin extends along the path of the atmospheric river discussed in the previous section, in a band stretching from the strait of Gibraltar all the way to the Alps, covering most of the eastern half of the Iberian Peninsula and southeast France. In all these regions, moisture from the North Atlantic is also a significant source, but tropical and subtropical water

vapor is clearly the most important contribution. In the north-eastern tip of the Iberian Peninsula and southeast France there is a relevant additional input from Western Mediterranean humidity (Fig. 12a), and in the French Massif Central, even modest precipitation amounts from Central Mediterranean evaporation (Fig. 12b). These areas where all major source contributions overlap are precisely the most impacted by the event and where the highest rainfall accumulations were recorded.

Table 2 shows the area averaged relative contribution of each source over northeast Spain and southeast France (region number 2, outlined in red in Fig. 2b, the same used for the vertical distribution of moisture analysis in Fig. 11). In this region, which includes the Pyrenees and the French Massif Central mountains where the most intense downpours occurred, tropical and subtropical sources are clearly dominant, with a contribution surpassing 50%. Western Mediterranean and North Atlantic moisture play an intermediate role, contributing each between 15 and 20%. Of the latter two sources, North Atlantic water

vapor is more relevant in the Pyrenees whereas that of the Western Mediterranean is so in the Massif Central. The input of the Central Mediterranean is on average negligible, of only around 3%. These results indicate that in the most affected areas, the contribution to precipitation from remote sources (about 70%) is much more important than that from local sources (less than 20%). The residual amount (11.8%) is, as in the October event, a sum of small contributions from other various sources. We note, however, that although the share of Western Mediterranean moisture is somewhat modest, its relevance is particularly

noteworthy; Fig. 12 suggests that without a contribution from the Mediterranean, rainfall accumulations in northeast Spain and southeast France would be comparable to those in many other regions of the Iberian Peninsula, and it is likely that the damage caused would have been much less.

**Table 2.** Same as Table 1 but from November 6 at 06:00 UTC to November 9 at 06:00 UTC and over region 2 in Fig. 2b.

| | Western Mediterranean | Central Mediterranean | North Atlantic | Tropical and Subtropical |
|---|---|---|---|---|
| **Relative Contribution (%)** | 15,60 | 2,96 | 18,20 | 51,39 |

## 5   Summary and conclusions

Torrential rain episodes causing flooding are recurrent features of climate on the shores of the Western Mediterranean. The meteorological drivers for such events can be quite different and nevertheless result in similar outcomes, with catastrophic consequences in terms of damages. We investigate here this type of episodes on the basis of a common hypothesis; for the most extreme events occur, one of the necessary ingredients is a large amount of precipitable water, which is to a great extent advected from remote regions.

We selected two infamous Western Mediterranean high precipitation events occurred during the same season, autumn of 1982 (October and November). Both evolved from very different synoptic situations. The case of October was more thermally driven, with the presence of cold air aloft associated with an upper level cut-off low, and deep convection developing and organizing in the form of a mesoscale convective system. In contrast, the November case was more dynamically forced, since it unfolded in the prefrontal and frontal zone of a strong Atlantic baroclinic storm. In this event, orography played a very relevant role, by enhancing the ascent producing precipitation and, in some mountain ranges such as the Pyrenees, also by triggering deep convection. The configurations of the selected cases represent two of the most frequently found during these episodes.

To assess the relevance of locally generated and remote precipitable water, we analyzed four potential moisture sources: evaporation in the Western or Central Mediterranean, evaporation in the North Atlantic and advection from the tropics and subtropics. Mediterranean sources were regarded as local while tropical, subtropical and Atlantic sources were considered as remote sources. Simulations were carried out with the WRF atmospheric model coupled with a moisture tagging technique, the so-called WRF-WVT tool. Lateral boundary forcing came from ERA-Interim reanalysis and a single domain at 20 km resolution was used for calculations. In addition to estimating the contribution of the different sources to the large rainfall accumulations recorded during the episodes, we analyzed the vertical distribution of moisture transport toward the affected areas, in order to obtain a three dimensional diagnosis of the involvement of water vapor from each source in the dynamics of the events. As a result of our findings, we state the following conclusions.

- In both episodes, the largest moisture contribution to the torrential rains was from tropical and subtropical sources. In the case of November, more than half of the rainfall recorded in the most affected area came from this origin, while in the case of October its predominance was somewhat less pronounced, representing around 31% of the total rainfall.

- In the October event, evaporated moisture in the Western and Central Mediterranean, i.e. local moisture, played a very important role, with these sources contributing nearly 20% of total precipitation each. Evaporated moisture in the North Atlantic was also a significant contributor, accounting for around 15% of total precipitation, although it was the least important of the four sources.

- In the November event, the North Atlantic and the Western Mediterranean acted as secondary sources, while the contribution of the Central Mediterranean was almost negligible. Even so, the Mediterranean's contribution is particularly noteworthy: many regions in the Iberian Peninsula received large amounts of rain, coming from Atlantic and tropical and

subtropical moisture sources; however, the extra input from the Mediterranean in northeast Spain and southeast France caused the rainfall in these areas to be even higher, so they ultimately were the most damaged areas.

- As for the distinction between remote and local sources, in the October event the contribution of both was similar whereas in the November case the largest share was clearly from remote sources.

- Moisture transport at medium and high levels played a key role in producing the observed large amounts of rainfall. Most water vapor at these layers resulted from long distance advection from the tropics and subtropics, which, as mentioned above, was the main source for the extreme precipitation. There were also high mixing ratios from this remote origin at lower layers, but the maximum values were at medium levels of the atmosphere.

- In the lower layers of the atmosphere, moisture was generally mostly from local evaporative sources in the Western and Central Mediterranean, while water vapour from evaporation in the North Atlantic was distributed at different levels.

- In both cases, moisture from the tropics and subtropics was transported through very defined moisture plumes or atmospheric rivers.

- The combination of high water vapor content at low levels from local sources and at middle and upper levels from remote sources yielded very large values of total precipitable vapor in the column in both events, but more so in the October case.

Our results suggest that the role played by remote sources is fundamental in producing the extraordinary rain accumulations observed in this type of extreme events and that the contribution of local Mediterranean sources is not sufficient to reach such high values. To corroborate the idea that remote sources of moisture from the tropics contribute to an important fraction of extreme precipitaiton events in the midlatitudes, many more episodes should be analysed. In this sense, this work is intended as a first step in applying the water vapor tracer method to many other cases in order to obtain more robust conclusions.

*Competing interests.* Authors declare that no competing interests are present.

*Acknowledgements.* Funding comes from the Spanish Ministerio de Economia y Competitividad OPERMO (CGL2017-89859-R to GMM and DIC) and M-CostAdapt (CTM2017-83655-C2-2-R to MCLL) projects, the European Union Interreg V POCTEFA project (EFA210/16 PIRAGUA to MCLL) and the CRETUS strategic partnership (AGRUP2015/02 to GMM and DIC). All these programs are co-funded by the European Union ERDF. ERA-Interim data was provided by ECMWF. Computation took place at CESGA (Centro de Supercomputacion de Galicia), Santiago de Compostela, Galicia, Spain. We would like to thank Oreste Reale and an anonymous reviewer for their suggestions and comments, which helped to improve the manuscript. This research is part of the HyMeX international programme.

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
