# Peer review of "Local and remote moisture sources for extreme precipitation: a study of the two catastrophic 1982 Western Mediterranean episodes"

_Hydrology and Earth System Sciences, 2018_

## Referee Comment (RC1) · Anonymous Referee #1 · 5 Dec 2018

This paper put the interest in one of the topics within the last times: the link between the origin of moisture and the occurrence of precipitation. Although this reviewer has many important general comments, the paper seems interesting to me (and for the scientific community), and after being improved I will recommend its publication.

In general:

The authors need to change the title because it is possible that these two events are "famous" in Spain, or in the Iberian Peninsula, but not in the international community.

[Figure]

Please change "famous", or delete it.

Both selected case occurred during the same year, 1982, why the authors select the events only this year? It is impossible that there is no other case in another year. Please see the work by Ramos et al (2017; DOI: 10.1002/joc.4726) where a ranking of events where done; or the important heavy rains in Lisbon during 26 October 2006 [https://earthobservatory.nasa.gov/images/17545/floods-in-portugal] with associated floods, and there are other examples.

So, the authors need to clarify this fact, and justify comparing with other extreme events the selection of these particular cases, because the event during October 1982 does not appears in the 10 first ranked events of high precipitation in Ramos et al (2017) and the November one appears in 3rd and 5th position for events with 3 days in duration.

The methodology about the moisture attribution is based on the WVT method and the WRF-WVT tool. Nowadays it seems to be a great tool and corroborated in different paper and applications, but the experiments are impossible to check or prove, as the model is nor freely available for the scientific community. The developer, one of the authors, needs to think in this option as many journals requires accessibility to the software and capability of others future authors to repeat the experiments. This is only a comment.

Another comment about the references cited in the paper. There is a copious quantity of self-references (1/5), and in two cases there are in Spanish. That is the case for those related with "gota fria" and a 1987 Ph.D. thesis. Both references are used to cite the synoptic meteorological system known as cut-off-lows (COLs). Checking the literature about them, there is a special issue published in Meteorological and Atmospheric Physics (MAP) journal in 2007, and no one of the papers included in this compendium were referenced. See in https://link.springer.com/journal/703/96/1/page/1. It would be excellent that the authors take a look at those papers concerning, at least, the Iberian Peninsula. The papers cited they have already been sufficiently amortized.

On the other hand, there are two main papers related to the characteristic of the COLs published after the both cited (one in 1987 and the thesis 1991, and in Spanish): Climatological features of Cut-off low systems in the Northern Hemisphere in Journal of Climate (2005) [https://doi.org/10.1175/JCLI3386.1] and Identification and Climatology of COLs near the Tropopause in Annals of the New York Academy of Sciences in 2008 [doi: 10.1196/annals.1446.016].

Although the paper is focused in the western Mediterranean region, the role of the COLs systems is necessary to put in a global context, showing that these systems occur over other regions around the world causing similar amounts of precipitation or if the effects are also important as in the Mediterranean area.

In the second page, the authors raised a number of questions and they listed five papers using different methodologies. Again, checking the newest literature there are other methods (and I will not go into isotopes methodology) not included here. Lagrangian approaches using backward techniques to follow changes in moisture were used to identify moisture transport from a global point of view and at regional scale during the last year, and it is highlighted the papers by Gimeno et al. (2010, 2011, 2012, 2013). The review paper about the "Oceanic and Terrestrial Sources of Continental Precipitation" is nowadays a seminal reference in this topic. The author should also not forget some papers using other models that justify the contribution of moisture to extreme events, like those by:

Sodemann, H. & Zubler, E. Seasonal and inter-annual variability of the moisture sources for Alpine precipitation during 1995–2002. Int. J. Clim. 2010, 30, 947–961

Schicker, I. et al. Origin and transport of Mediterranean moisture and air. Atmos. Chem. Phys. 2010, 10, 5089–5105.

Ciric, D. et al. Wet Spells and Associated Moisture Sources Anomalies across Danube River Basin. Water 2017, 9, 615. Liberato et al. (2013) Moisture Sources and Large‐Scale Dynamics Associated With a Flash Flood Event. In Lagrangian Mod-
eling of the Atmosphere, Volume 200. Book Series: Geophysical Monograph Series Or those related to synoptic conditions, for instance:

Pfahl, S. Characterising the relationship between weather extremes in Europe and synoptic circulation features. Nat. Hazards Earth Syst. Sci. 2014, 14, 1461–1475.

Dayan et al (2015). Review Article: Atmospheric conditions inducing extreme precipitation over the eastern and western Mediterranean. NHESS. doi:10.5194/nhess-15-2525-2015

And this review assumes that there are many lacks in the references included in this review. So, the authors of the paper, need to improve the list reference as it is evident that in the present manuscript important references are still lacking to put in context the problematic.

About the definition of the "predefined" source of moisture: this is from my point of view the major point to check in the paper. It has no sense define all the Northern Atlantic or all the Tropical Atlantic areas. There tools to detect the specific sources of moisture for both events, and then use the WRF-WVT tool. Recently in a discussion paper in ESD https://www.earth-syst-dynam-discuss.net/esd-2018-76/, one of the authors use a Lagrangian methodology to define it. So, why not in this paper? If the definition is more properly the results will be more justifiable. If the authors do not redefine the limits of the sources, they need to justify better this fact in the actual manuscript. On the other hand, there are some papers that analyze the moisture transport for the Iberian Peninsula: "Where Does the Iberian Peninsula Moisture Come from? An Answer Based on a Lagrangian Approach". J. Hydrometeorol. 2010, 11, 421–436 in which a regionalization was done.

Specific comments:

Page 2, line 5 and line 30: the authors say that the "moisture as a key factor is often undervalued or not considered in depth" in line 5, and then in line 30 affirm that the

cited papers "have provided quite a detail knowledge about the origin of the moisture feeding extreme rainfall ...". This does not make sense, or yes or not. They should consider that the sentence in line 5 is to hard. Please, rewrite it. There many works about the role of the moisture for extreme precipitation.

Page 3, line 3: the application of this tool is not a "novelty". The same authors have many papers using this technique, including researches about extreme precipitation related to Atmospheric Rivers. Of course, in this paper the meteorological systems analyzed are not over the same regions (Atlantic and Pacific). But they have at least five or six papers (or more) using this tool.

Page 3 line 11: add a reference about the validation of the tool.

Page 3 line 25: which are the common types of situation associated with HPEs in the region? Clarify in this part of the text.

Figure 1: the resolution is not good in this version.

Page 5, line 6-11: the 2d or 3D definition needs more explanation. Why the Arabian Sea could influence the ST source? This needs also justification. This review assumes that the Gulf of Mexico affect the Iberian Peninsula, as many works affirm (and they need to be cited here, of course).

Page 5 line 7: why this division of the Mediterranean Sea? Please add a reference. It seems an usually division used in previous studies of the Mediterranean Seas as sources of moisture: e.g. Nieto et al., 2010 or Schicker et al., 2010 based in works of Millan et al 1997 and 2002

R. Nieto, L. Gimeno, A. Drumond, E. Hernández. 2010. A Lagrangian identification of the main moisture sources and sinks affecting the Mediterranean area. WSEAS Trans. Environ. Dev. 6 (5), 365-374

I. Schicker, S. Radanovics, P. Seibert. 2010. Origin and transport of Mediterranean moisture and air. Atmos. Chem. Phys., 10, 5089-5105, 10.5194/acp-10-5089-2010.

[Figure]

M. Millan, M.J. Sanz, R. Salvador, E. Mantilla. 2002. Atmospheric dynamics and ozone cycles related to nitrogen deposition in the western Mediterranean. Environ. Pollut., 118, 167-186, 10.1016/S0269-7491(01)00311-6

Page 5, line 14: why the authors ignore the continental areas as sources of moisture? They assume, but they need to justify this based on other paper(s).

Page 5 line 16: I assume that these 10 days are used because this time is the typical definition of the mean time-averaged lifetime of water vapor in the atmosphere. Many many papers using this time cited Numaguti et al (1999): Origin and recycling processes of precipitating water over the Eurasian continent: Experiments using an atmospheric general circulation model. J. Geophys. Res., 104 (D2), 1957-1972, 10.1029/1998JD200026.

How sensitive are the results to this time used? In a recent paper by Läderach and Sodemann (2016) they obtained times as short as 4 - 5 days. Did the authors check it?

Läderach & Sodemann, 2016 GRL 43(2), 924-933, doi:10.1002/2015GL067449

Figure 2: the vertical scale in b): is it the elevation. Put this in the caption, please.

Page 6, line 2: why the authors span the experiment to 12 days?

Page 6, line 14: which is MESCAN? It is not defined previously.

In general for the synoptic configuration. It is needed plots with the field using not only WRF outputs.

Page 6, line 18-20: if this day occurs a COL, please add the geopotential field at 200 hPa, or at least, 300 hPa, to show the low in high levels. To show the instability add a field of convergence.

Page 6, line 6: this low referred here is the Cut-off low. And perhaps is not this low the system that stopped the flow, but rather the anticyclone itself and the low pressure

system over Iceland that pick the moisture to the north, as it is also evident in Fig5c, where an Atmospheric river associated with it is clear.

Figure 5 (and fig. 10): it could be useful a VIMF field to see the prevalence of the flux.

Page 10, line 5: the 20% moisture misprice has a similar contribution than those from Central Mediterranean or more... Why consider CM and no other sources, for instance the continental ones? Or why the authors do not consider to join CM and WM?

Page 11. How the authors could difference from which part of the ST source the final moisture for precipitation comes from? Because the ST contains also moisture that is advected by the AR to northern latitudes. So it seems that the moisture comes only from the eastern part of the Atlantic Ocean of the ST, and over the Sahel region, that it is not completely taken into account (the plot only show a box from 7.5°N. Could be the source of moisture even further south?. That is the problem if the sources were previously predefined.

Page 11, line20: the omega pattern is typical in a cut-off low formation. The October case appears as a pure cut-off low, but it needs to come from an elongated trough, that develops to a phase of tear-off (when an omega configuration is normal. So the big difference between this November case and the October one is that the COL, in this case, is bigger in size.

Page 12: the affected areas in relation to the cut-off low position were analyzed in a paper included in the special issue in MAP journal commented previously.

Page 14: after these results I recommend the authors to joint CM and WM.

---

## Referee Comment (RC2) · Reale (Referee) · 4 Jan 2019

This is an excellent piece of research that brings a substantial advance to the complex issue of moisture sources related to flood-producing precipitation events over the Mediterranean region. While the writing could be improved, the results are very convincing. I find the methodology particularly praiseworthy. As such, I recommend the article to be accepted after some minor revision.

General comment: expand the focus.

I agree with the other reviewer that the Authors should consider changing the title. Aside from defining as 'famous' events that may not be known outside the hydrology, engineering, and meteorology communities in Spain, I would rather use the term 'infamous' to describe catastrophic events that have caused death and destruction. Even better, I would avoid 'fame' entirely and perhaps refer to the events as 'catastrophic'. The term 'famous' also appears in page 17, second par.

Most important, I suggest to modify the introduction, in order to provide a broader motivation that can make the article relevant to a much larger community. The description of general mechanisms as it is cannot provide objective, generic, absolute 'causes'. Furthermore, the Authors themselves acknowledge that the 2 events are very different one from the other. Therefore, I suggest to broaden the focus of this article, by connecting this work with other research.

Conditions for the development of these events surely are strong instability, presence of some circulation that organizes the flow, exploiting orographic contribution. However, the puzzling aspect is that most of these conditions, for example Mediterranean baroclinic cyclones, are often present but are rarely associated with extreme precipitation. It is only a very small subset of Mediterranean cyclones that cause catastrophic events. Furthermore, in some instance the Mediterranean cyclone could be less relevant than the large-scale southerly flow associated with larger cyclonic circulations outside the Mediterranean. So, the Authors may consider starting with the statement that the presence (or absence) of intense moisture transport anomalies on a very large scale could be the critical, discriminating factor between many situations apparently similar but in which only one produces an extreme precipitation event.

Another suggestion is to think in a more 'global' scale. The Authors' experiments, unlike previous work, are both at very high resolution and encompass a very large domain. As such, they have the possibility of linking Mediterranean floods with the global scale. If this is done, the article will attract a much larger set of readers interested in the subject of tropical-extratropical connections.

As a starting work, consider the final part of Wu et al. (2013). In that work, we linked 2 cases of flood-producing precipitation over Europe with the so-called 6-9 day African Easterly Waves (for the part of the article relevant to your work, please see Section 3d, from page 6765 onwards). The 6-9 waves are different from the well-known 3-6 day waves, because they form to the north of the African Easterly Jet, at the jet level (about 600 hPa) and they travel northward. They can be conceived as a 'relaxation' in the subtropical high pressure that bridges mid latitude low pressure systems with the Inter Tropical Convergence Zone (ITCZ).

From this article's perspective, 6-9 day waves are relevant because they represent a way of connecting tropical moisture generated within the ITCZ with midlatitude systems. If one of such waves acts in phase with a deep midlatitude cyclone, a stream of moisture can leave the ITCZ, travel in a relatively stable area associated with the relaxed subtropical high (and thus without loosing moisture) and 'connect' with the warm advection ahead of a frontal system in the midlatitudes. Then, any mechanism able to concentrate and release this enormous amount of moisture over a small area, can cause a flood-producing precipitation event. Figures 14, 15, and 16 from Wu et al. (2013) illustrate this aspect for extreme precipitation events occurred in 2000 and 2002, respectively. It seems that the plume of moisture associated with Fig.3b, Fig. 5d, in this work, bears remarkable similarities.

Most relevant for this work are also the papers by Knippertz (2003), Knippertz et al. (2003) which connect episodes of extreme precipiation over northwest Africa with anomalous advection from the tropics, and place these into the context of a tropical-extratropical interaction.

Schepanksi and Knippertz (2011) further expand in this direction and finds in the Soudano-Saharan depression a key element connecting tropics with midlatides. We think that all these results are very consistent with each other, and simply focus on different aspects of the moisture transport.

[Figure]

It is important to notice that anomalous moisture advection from the tropical Atlantic has been noted also outside the Mediterranean region: the study of Stohl et al. (2008) identifies a very similar moisture path for precipitation events in Norway, connecting these with the tropical Atlantic.

Minor comments:

The WVT could become a formidable tool, particularly because is coupled with the WRF, which is very well known and used worldwide. The Authors should consider distributing it, either through their own portal, or in collaboration with an WRF development team. It would gather widespread attention if it became an easily accessible methodology. I find particularly important, compared to earlier studies, the ability of investigating sources on a 3D scale.

Figures. The clarity of the figures illustrating the synoptic situation could be improved. Even coastlines and geographic features are barely detectable. In Figure 3a, I suggest blanking out the temperatures around 0C (such as -2 2) so as to have a white strip between cold and warmer air, and use less intense colors. If done in GrADS, the Authors could consider the use of transparencies which is now an option and allows more readable plots. Otherwise, just use lighter colors.

In figure 3b, I suggest to blank out completely the values of tpw less than 10mm. These are not relevant to this work, indicate simply drier air, and by eliminating them the emphasis would be given to the huge moisture plume stretching from the subtropical Atlantic towards the Iberian peninusula.

Similar suggestions for Figure 8.

In the text, there are some sentences whose clarity could be improved. See for example page 4, lines 7-8, d

Page 18, line 14.. a semantic issue.. Instead of 'verify the hypothesis' .. . Consider something like 'corroborate the idea that remote sources of moisture from the tropics

contribute to an important fraction of extreme precipitaiton events in the midlatitudes..'

In summary, aside from these relatively minor suggestions, I believe that the article is a great contribution to precipitation research, and I recommend acceptance after minor revision. However, I believe that the article would benefit by placing the results into the broader context of tropical-extratropical interactions and large scale advection of tropical moisture.

Oreste Reale

References

Knippertz, P., 2003: Tropical–extratropical interactions causing precipitation in northwest Africa: Statistical analysis and seasonal variations. Mon. Wea. Rev., 131, 3069–3076.

Knippertz, A. H. Fink,A.Reiner, and P. Speth, 2003: Three late summer/early autumn cases of tropical–extratropical interactions causing precipitation in northwest Africa. Mon. Wea. Rev., 131, 116–135.

Schepanski, K., and P. Knippertz, 2011: Soudano-Saharan depressions and their importance for precipitation and dust: A new perspective on a classical synoptic concept. Quart. J. Roy. Meteor. Soc., 137, 1431–1445.

Stohl, A., C. Forster, and H. Sodemann, 2008: Remote sources of water vapor forming precipitation on the Norwegian west coast at 608N—A tale of hurricanes and an atmospheric river. J. Geophys. Res., 113, D05102, doi:10.1029/2007JD009006.

Wu, M.-L, O. Reale, S. Schubert, 2013: A characterization of African Easterly Waves on 2.5-6 and 6-9 day time scales. Journal of Climate, 26, 6750-6774.

---

## Author Comment (AC1) · 1 Mar 2019

This paper put the interest in one of the topics within the last times: the link between the origin of moisture and the occurrence of precipitation. Although this reviewer has many important general comments, the paper seems interesting to me (and for the scientific community), and after being improved I will recommend its publication.

Thank you very much for your review. We believe that the modifications you suggest will improve the manuscript.  Please, find below the responses to your comments.

In general:

The authors need to change the title because it is possible that these two events are "famous" in Spain, or in the Iberian Peninsula, but not in the international community.

Please change "famous", or delete it.

We agree with the reviewer, so the word "famous" is going to be replaced by "catastrophic".

Both selected case occurred during the same year, 1982, why the authors select the events only this year? It is impossible that there is no other case in another year. Please see the work by Ramos et al (2017; DOI: 10.1002/joc.4726) where a ranking of events where done; or the important heavy rains in Lisbon during 26 October 2006 [https://earthobservatory.nasa.gov/images/17545/floods-in-portugal] with associated floods, and there are other examples.

So, the authors need to clarify this fact, and justify comparing with other extreme events the selection of these particular cases, because the event during October 1982 does not appears in the 10 first ranked events of high precipitation in Ramos et al (2017) and the November one appears in 3rd and 5th position for events with 3 days in duration.

Several factors support the choice of these two events. Both events appear, for example, in the list of major flood disasters in Europe between 1950 and 2005 (Barredo, 2007). As discussed in the article, both events exceeded the death toll of 40, making them two of the most catastrophic events in the Mediterranean region in the second half of the 20th century. The amounts of rain recorded were sensational; in addition to those cited in the paper, some researchers have gone so far as to ensure that in the October event more than 1000 mm could have fallen in 15 hours in the observatory located in La Muela de Cortes de Pallás (Valencia), which would be a historical record in Spain. Finally, the fact that the two events occurred in such a short time period and triggered by such different weather conditions (but both very common in this type of episodes) makes them more interesting.

As for the Ramos et al. (2017) method, it takes into account the accumulated precipitation in periods of 3, 5, 7 or 10 days and in the entire Iberian Peninsula (added area of all points with precipitation anomalies greater than two standard deviations), so that events with very high amounts of rainfall in a short time and in more localized regions tend to lose importance in this

ranking, despite being the ones that usually cause more damage (such as the October 1982 event).Following the referee's recommendation, we will add a clarification on the reason for the choice of these episodes in the fifth paragraph of the introduction (L14-L27 P3).

Barredo, J. I.: Major flood disasters in Europe: 1950-2005, Natural Hazards, 42, 125–148, https://doi.org/10.1007/s11069-006-9065-2, 2007.

The methodology about the moisture attribution is based on the WVT method and the WRF-WVT tool. Nowadays it seems to be a great tool and corroborated in different paper and applications, but the experiments are impossible to check or prove, as the model is nor freely available for the scientific community. The developer, one of the authors, needs to think in this option as many journals requires accessibility to the software and capability of others future authors to repeat the experiments. This is only a comment.

Thank you for your suggestion. We are currently considering making the code open access.

Another comment about the references cited in the paper. There is a copious quantity of self-references (1/5), and in two cases there are in Spanish. That is the case for those related with "gota fria" and a 1987 Ph.D. thesis. Both references are used to cite the synoptic meteorological system known as cut-off-lows (COLs). Checking the literature about them, there is a special issue published in Meteorological and Atmospheric Physics (MAP) journal in 2007, and no one of the papers included in this compendium were referenced. See in https://link.springer.com/journal/703/96/1/page/1. It would be excellent that the authors take a look at those papers concerning, at least, the Iberian Peninsula. The papers cited they have already been sufficiently amortized.

On the other hand, there are two main papers related to the characteristic of the COLs published after the both cited (one in 1987 and the thesis 1991, and in Spanish): Climatological features of Cut-off low systems in the Northern Hemisphere in Journal of Climate (2005) [https://doi.org/10.1175/JCLI3386.1] and Identification and Climatology of COLs near the Tropopause in Annals of the New York Academy of Sciences in 2008 [doi: 10.1196/annals.1446.016].

Although the paper is focused in the western Mediterranean region, the role of the COLs systems is necessary to put in a global context, showing that these systems occur over other regions around the world causing similar amounts of precipitation or if the effects are also important as in the Mediterranean area.

We disagree with the reviewer in that there is a one to one relationship between the presence of cut off lows and extreme precipitation in the Mediterranean and much less so in other parts of the world. It is indeed true that the cold air aloft associated with COL systems enhances instability and favors convection and precipitation. But this does not mean that this configuration will always result in extreme rains, nor all extreme precipitation events with flooding are necessarily produced by a COL. The main point of the paper is to provide support to the hypothesis that the common feature in all extreme precipitation and catastrophic flooding cases in the Mediterranean is not the synoptic setting (cut off low or not) but a high amount of precipitable water. Of course, some lifting mechanism is also needed, but this does

not always have to be the same (the most common ones are nicely summarized in the review paper by Dayan et al., 2015, suggested by the reviewer in the next comment). Instability and convection can result from the presence of low level warm and moist air and not only from an upper level cold air mass, and air parcels can also ascend forced by some dynamical mechanism, such as a frontal circulation or upper level divergence. We illustrate this idea very clearly with the two examples we chose for the study, which are very different in terms of synoptic forcing, as it is discussed in the text.

The equivalence between the presence of a cut off low and extreme precipitation and flooding in the Mediterranean is a common misconception, that in places like Spain has even been embraced by the general public and permeated to everyday language, making cut-off low ("gota fría") and extreme rain and flooding synonymous. This is the main reason why we purposely avoided discussing cut off lows in the text. We also provided only brief information on any other specific synoptic setting that can produce HPEs in the Mediterranean, shifting the focus entirely to moisture, which is the main topic of the article. However, given the frequent involvement of COLs in these extreme events, we will add the two suggested references (Nieto et al., 2005, 2008) when COLs are mentioned in the introduction (P2 L4), as per the reviewer's request.

The references to Llasat (1987, 1991) are justified by the fact that there is scarce literature directly dealing with these particular cases. We do not refer to these articles in allusion to cut-off-lows (COLs) specifically, but to allude to a more in depth general meteorological analysis of the cases in our study.

In the second page, the authors raised a number of questions and they listed five papers using different methodologies. Again, checking the newest literature there are other methods (and I will not go into isotopes methodology) not included here. Lagrangian approaches using backward techniques to follow changes in moisture were used to identify moisture transport from a global point of view and at regional scale during the last year, and it is highlighted the papers by Gimeno et al. (2010, 2011, 2012, 2013). The review paper about the "Oceanic and Terrestrial Sources of Continental Precipitation" is nowadays a seminal reference in this topic. The author should also not forget some papers using other models that justify the contribution of moisture to extreme events, like those by:

 Sodemann, H. & Zubler, E. Seasonal and inter-annual variability of the moisture sources for Alpine precipitation during 1995–2002. Int. J. Clim. 2010, 30, 947–961

Schicker, I. et al. Origin and transport of Mediterranean moisture and air. Atmos. Chem. Phys. 2010, 10, 5089–5105.

Ciric, D. et al. Wet Spells and Associated Moisture Sources Anomalies across Danube River Basin. Water 2017, 9, 615. Liberato et al. (2013) Moisture Sources and Large  RScale Dynamics Associated With a Flash Flood Event. In Lagrangian Modeling of the Atmosphere, Volume 200. Book Series: Geophysical Monograph Series

Or those related to synoptic conditions, for instance:

Pfahl, S. Characterising the relationship between weather extremes in Europe and synoptic circulation features. Nat. Hazards Earth Syst. Sci. 2014, 14, 1461–1475.

Dayan et al (2015). Review Article: Atmospheric conditions inducing extreme precipitation over the eastern and western Mediterranean. NHESS. doi:10.5194/nhess-15- 2525-2015

And this review assumes that there are many lacks in the references included in this review. So, the authors of the paper, need to improve the list reference as it is evident that in the present manuscript important references are still lacking to put in context the problematic.

In the third paragraph of the introduction (L13 P2- L2 P3), we provide the reader with a general view of the state of the art in the research on moisture sources for extreme precipitation in the Mediterranean. With this purpose, we referenced six authors who used Lagrangian methods, the most commonly employed, and one author (the only one we know of) who uses the tracer's method, just like us. We believe that these references are sufficient to introduce the reader to the numerical study of moisture sources for extreme precipitation in the Mediterranean. Obviously, these methods have also been used for the study of extreme precipitation events in other regions of the planet, as well as for the study of moisture sources, both in the Mediterranean and in other regions in the climatic sense. But we do not believe that it is necessary in this paper to provide the reader with such a global vision of the field of moisture tracking. In a previous article (Insua-Costa and Miguez-Macho, 2018), which is cited in the text, we did a more general review of the state of the art of numerical methods for the study of moisture sources, but we do not think it is pertinent to repeat it here. Notwithstanding, a reference to Gimeno et al. (2012) on line 13 of page 2 does seem appropriate because, as the referee mentions, it is nowadays a seminal reference in this topic. We are going to add it.

Likewise, in our brief synoptic analysis of the events, we believe that it is enough to reference those publications analyzing the weather conditions of the case studies chosen for this paper and not others that deal with the problem of extreme rainfall in general. However, the review paper by Dayan et al. (2015) is very suitable as a reference in the first part of the introduction, when the most common settings for extreme precipitation in the western Mediterranean are mentioned. We will now include it and we thank the reviewer for the suggestion.

Insua-Costa, D. and Miguez-Macho, G.: A new moisture tagging capability in the Weather Research and Forecasting model: Formulation, validation and application to the 2014 Great Lake-effect snowstorm, Earth System Dynamics, 9, 167–185, https://doi.org/10.5194/esd-9-167-2018, 2018.

Gimeno, L., Stohl, A., Trigo, R. M., Dominguez, F., Yoshimura, K., Yu, L., Drumond, A., Durn-Quesada, A. M., and Nieto, R.: Oceanic and terrestrial sources of continental precipitation, Rev. Geophys., 50, 1–41, https://doi.org/10.1029/2012RG000389, 2012.

About the definition of the "predefined" source of moisture: this is from my point of view the major point to check in the paper. It has no sense define all the Northern Atlantic or all the Tropical Atlantic areas. There tools to detect the specific sources of moisture for both events, and then use the WRF-WVT tool. Recently in a discussion paper in ESD https://www.earth-systdynam-discuss.net/esd-2018-76/, one of the authors use a Lagrangian methodology to define it. So, why not in this paper? If the definition is more properly the results will be more justifiable. If the authors do not redefine the limits of the sources, they need to justify better this fact in the actual manuscript. On the other hand, there are some papers that analyze the moisture transport for the Iberian Peninsula: "Where Does the Iberian Peninsula Moisture Come from? An Answer Based on a Lagrangian Approach". J. Hydrometeorol. 2010, 11, 421–436 in which a regionalization was done.

We fully agree with the reviewer in that this is a controversial point of the article. In fact, at first we ourselves also thought that the best way to select the sources was the one proposed by the reviewer (like in https://www.earth-syst-dynam-discuss.net/esd-2018-76). However, bearing in mind that our long-term goal is to apply the method to a much larger number of events (as commented in the paper L14-L15 P18), in order to construct a climatology of moisture sources for extreme events, the targeting of originating regions cannot be done following the aforementioned strategy. This is because, with the proposed method, different sources would be chosen for each event, making it difficult to calculate a final average. The selected sources have to always be the same. For instance, to map the spatial distribution of the average moisture input of the tropics to precipitation in these episodes, one would have to divide the general tropical region into smaller subareas down to the desired resolution, and then run the model for each of them and for all cases to finally obtain the average contribution in such detail. For 1x1 degree squares, this means hundreds of simulations just for one case. The selection proposed here is based on the choice of quite extensive sources, which does not mean they are not enlightening: a distinction is made between local (Mediterranean) and remote (Atlantic) humidity; within the remote we distinguish between tropical and non-tropical and within the local between Western and Central Mediterranean. Furthermore, the main point of the paper is precisely to find out whether moisture is of remote or local origin, and not the detailed geographic location of the source.

For each selected moisture source a simulation must be carried out, with the corresponding increase in computational cost, so it would not be feasible to increase the number of selected sources much more if a large number of cases (in the hundreds) is to be analysed in the future. This is why we chose sources as extensive as the entire North Atlantic or the entire Tropical and Subtropical Atlantic.

We agree with the reviewer in that the reasoning behind our source selection should be made clearer, so we will add further discussion in section "2.2 Experimental design".

Specific comments:

Page 2, line 5 and line 30: the authors say that the "moisture as a key factor is often undervalued or not considered in depth" in line 5, and then in line 30 affirm that the cited papers "have provided quite a detail knowledge about the origin of the moisture feeding extreme rainfall ...". This does not make sense, or yes or not. They should consider that the sentence in line 5 is to hard. Please, rewrite it. There many works about the role of the moisture for extreme precipitation.

We are aware that there is research focused on moisture in these events in the literature (see the citations in the introduction section), but we think that other aspects related to Mediterranean extreme rainfall have been much more studied. It is not uncommon to find articles on the causes of some event of these characteristics in which the role of humidity is considered totally secondary. In addition, we also believe that moisture as a key factor is often not sufficiently taken into account in the warning systems and forecasts of meteorological agencies.

However, in order to make this statement less controversial, we are going to change "is often" to "is sometimes".

Page 3, line 3: the application of this tool is not a "novelty". The same authors have many papers using this technique, including researches about extreme precipitation related to Atmospheric Rivers. Of course, in this paper the meteorological systems analyzed are not over the same regions (Atlantic and Pacific). But they have at least five or six papers (or more) using this tool.

The novelty lies in the fact that almost no other research (only one author) has used the Eulerian tracer method for the study of extreme precipitation events in the Mediterranean. All other authors used Lagrangian methods, as it is clearly discussed in the Introduction section. The application of the WRF-WVT tool in itself is of course no longer a novelty.

Page 3 line 11: add a reference about the validation of the tool.

Following the reviewer's suggestion, the reference from line 10 will be moved to line 11.

Page 3 line 25: which are the common types of situation associated with HPEs in the region? Clarify in this part of the text.

We agree so we are going to replace:

"A notable feature of these two episodes is that they represent two of the most common types of situation associated with HPEs in the NWMR, so the conclusions obtained in this work could be extrapolated to many other cases."

With:

"A notable feature of these two episodes is that they represent the two most common atmospheric circulation patterns associated with HPEs in the NWMR (see AP3 and AP13 weather types in the Romero et al., 1999b classification), so the conclusions obtained in this work could be extrapolated to many other cases."

Romero, R., Sumner, G., Ramis, C., & Genovés, A. (1999). A classification of the atmospheric circulation patterns producing significant daily rainfall in the Spanish Mediterranean area. International Journal of Climatology: A Journal of the Royal Meteorological Society, 19(7), 765-785.

Figure 1: the resolution is not good in this version.

The figure resolution will be improved.

Page 5, line 6-11: the 2d or 3D definition needs more explanation. Why the Arabian Sea could influence the ST source? This needs also justification. This review assumes that the Gulf of Mexico affect the Iberian Peninsula, as many works affirm (and they need to be cited here, of course).

Here, we were referring to the work of Krichak et al. (2015), discussed in the introduction section (page 3, L26), who found that tropical moisture exports from the Atlantic and the Arabian sea are involved in more than 50 intense heavy precipitation events in the Mediterranean region that they analyzed using reanalysis data.

We are aware that there are several studies showing the important role of the Gulf of Mexico as a moisture source for precipitation in the Iberian Peninsula. We did not reference those because our focus is specifically on the most intense heavy precipitation events only, and not on the sources for total annual or seasonal precipitation in general. However, we agree with the reviewer in that this general connection with the Gulf of Mexico should be mentioned, since heavy precipitation events account for a sizeable fraction of annual precipitation in several Mediterranean coastal areas, especially in Spain. We will now reference to Nieto et al. (2010) and Gimeno et al. (2012) in the introduction section.

Krichak, S. O., Barkan, J., Breitgand, J. S., Gualdi, S., and Feldstein, S. B.: The role of the export of tropical moisture into midlatitudes for extreme precipitation events in the Mediterranean region, Theoretical and Applied Climatology, 121, 499–515, https://doi.org/10.1007/s00704-014-1244-6, 2015.

R. Nieto, L. Gimeno, A. Drumond, E. Hernández. 2010. A Lagrangian identification of the main moisture sources and sinks affecting the Mediterranean area. WSEAS Trans. Environ. Dev. 6 (5), 365-374.

Gimeno, L., Stohl, A., Trigo, R. M., Dominguez, F., Yoshimura, K., Yu, L., Drumond, A., Durn-Quesada, A. M., and Nieto, R.: Oceanic and terrestrial sources of continental precipitation, Rev. Geophys., 50, 1–41, https://doi.org/10.1029/2012RG000389, 2012.

Page 5 line 7: why this division of the Mediterranean Sea? Please add a reference. It seems an usually division used in previous studies of the Mediterranean Seas as sources of moisture: e.g. Nieto et al., 2010 or Schicker et al., 2010 based in works of Millan et al 1997 and 2002

R. Nieto, L. Gimeno, A. Drumond, E. Hernández. 2010. A Lagrangian identification of the main moisture sources and sinks affecting the Mediterranean area. WSEAS Trans. Environ. Dev. 6 (5), 365-374

I. Schicker, S. Radanovics, P. Seibert. 2010. Origin and transport of Mediterranean moisture and air. Atmos. Chem. Phys., 10, 5089-5105, 10.5194/acp-10-5089-2010.

M. Millan, M.J. Sanz, R. Salvador, E. Mantilla. 2002. Atmospheric dynamics and ozone cycles related to nitrogen deposition in the western Mediterranean. Environ. Pollut., 118, 167-186, 10.1016/S0269-7491(01)00311-6

The division of the Mediterranean into Western and Central basins is based on common geographical criteria. There are many other authors that use the same or a very similar division in all fields of study and we do not think it is necessary to add any citation here.

Page 5, line 14: why the authors ignore the continental areas as sources of moisture? They assume, but they need to justify this based on other paper(s).

Evapotranspiration in continental Europe and Northwestern North America in late October and early November is rather little due to the cold temperatures and the natural cycle of vegetation. The contribution of the mostly desertic Northern Africa is safe to assume that is even smaller. Other possible continental sources in our domain in sub-Saharan Africa are included in the 3D tropical moisture source. However, as per the reviewer's request, we will better justify our assumption by adding a reference to Sodemann and Zubler (2010) and Drumond et al. (2011) at the end of the sentence "We assume that in autumn it is very diminished and hence it does not have a potentially important contribution". These moisture source studies explicitly find that autumn continental ET is not relevant for precipitation in regions around the Mediterranean.

Sodemann, H. and Zubler, E.: Seasonal and inter-annual variability of the moisture sources for Alpine precipitation during 1995– 2002, Int. J. Climatol., 30, 947–961, 2010.

Drumond, A., Nieto, R., Hernandez, E., and Gimeno, L.: A Lagrangian analysis of the variation in moisture sources related to drier and wetter conditions in regions around the Mediterranean Basin, Nat. Hazards Earth Syst. Sci., 11, 2307-2320, https://doi.org/10.5194/nhess-11-2307-2011, 2011.

Page 5 line 16: I assume that these 10 days are used because this time is the typical definition of the mean time-averaged lifetime of water vapor in the atmosphere. Many many papers using this time cited Numaguti et al (1999): Origin and recycling processes of precipitating water over the Eurasian continent: Experiments using an atmospheric general circulation model. J. Geophys. Res., 104 (D2), 1957-1972, 10.1029/1998JD200026.

How sensitive are the results to this time used? In a recent paper by Läderach and Sodemann (2016) they obtained times as short as 4 - 5 days. Did the authors check it?

Läderach & Sodemann, 2016 GRL 43(2), 924-933, doi:10.1002/2015GL067449

As stated in the paper "this 10-day period roughly coincides with the average residence time of water vapor in the atmosphere" and we refer to Trenberth (1998), which is the major reference in this subject, and a more recent study (Van Der Ent and Tuinenburg, 2017).

This later study confirms the traditional estimate of the average residence time of water vapor in the atmosphere (8-10 days), and explicitly rejects the conclusions of studies that suggest an average residence time of 4-5 days. In any case, our results would only be affected if the residence time were longer, not shorter, since the sooner we start the simulation, the lower the contribution of water vapor already present at initial time over the 2D sources to precipitation in the events. This initial water vapor does not have a known origin and we do not want it to interfere with our results.

Trenberth, K. E.: Atmospheric moisture residence times and cycling: Implications for rainfall rates and climate change, Climatic Change, 39, 667-694,https://doi.org/10.1023/A:1005319109110, 1998.

Van Der Ent, R. J. and Tuinenburg, O. A.: The residence time of water in the atmosphere revisited, Hydrology and Earth System Sciences, 21, 779–790, https://doi.org/10.5194/hess-21-779-2017, 2017

Figure 2: the vertical scale in b): is it the elevation. Put this in the caption, please.

The caption will be corrected as suggested.

Page 6, line 2: why the authors span the experiment to 12 days?

Please, see response to the previous comment on page 5. We start the simulations 10 days before the events invoking that this is the average time of water vapor in the atmosphere, precisely to assume that the contribution of initial moisture to precipitation is negligible on days 10, 11 and 12, which are the main day of the extreme precipitation event (day 11) and the days before and after. We analyze those 3 days in total.

Page 6, line 14: which is MESCAN? It is not defined previously.

We are going to replace the paragraph:

"Finally, for model validation we use the MESCAN precipitation analysis dataset (Soci et al., 2016), recently available in the ECMWF MARS (Meteorological Archival and Retrieval System) archive at 5.5 km resolution and covering our entire area of study."

With the following:

"Finally, for model precipitation validation we use the MESCAN (from Mesoscale Analysis; Soci et al., 2016) system, which combines a downscaled reanalysis and interpolated rain gauge measurements to get a high resolution (5.5 km) daily precipitation dataset. This product is recently available in the ECMWF MARS (Meteorological Archival and Retrieval System) and covers our entire area of study."

In general for the synoptic configuration. It is needed plots with the field using not only WRF outputs.

The use of the WRF model outputs to display the synoptic configuration is based on the use of spectral nudging in the simulations. Model results, except for moisture, are relaxed to ERA-Interim on the large-scale throughout the simulations and thus do not deviate substantially from reanalysis fields. It is practically equivalent to show one or the other.

Page 6, line 18-20: if this day occurs a COL, please add the geopotential field at 200 hPa, or at least, 300 hPa, to show the low in high levels. To show the instability add a field of convergence.

The figure below is the same as Figure 3 in the paper but with the geopotential height at 300 hPa instead of 500 hPa. The fields are very similar (disregarding their values). Therefore, the geopotential height at 500 hPa is sufficient to correctly show the COL.

[Figure]

As already discussed, the aim of this paper is not to describe in detail the synoptic or mesoscale configuration of the event. Thus, we do not think it is appropriate to go as far into this matter as to show the convergence field, because this would only distract the reader from the real goal of the article, which is to evidence that high amounts of moisture of mostly remote origin are involved in this kind of episodes. In addition, we believe that the article has already enough figures to enable readers to easily follow discussions and we do not want to overload it by adding more.

Page 6, line 6: this low referred here is the Cut-off low. And perhaps is not this low the system that stopped the flow, but rather the anticyclone itself and the low pressure system over Iceland that pick the moisture to the north, as it is also evident in Fig5c, where an Atmospheric river associated with it is clear.

Here we are referring to the low pressure system at low levels. As far as we know, a COL is only defined in the upper levels. We understand that this low at low levels appears as a result of the COL formation, but we wouldn't call it a COL.

When we say "The low pressure system situated over North Africa blocks the direct advance of evaporated moisture from the North Atlantic toward the Spanish Levant area" we are referring to the fact that this low organizes a flow that prevents North Atlantic humidity from directly reaching the affected area, and instead , it needs to go around the cyclone (traversing Africa) to do so.

Figure 5 (and fig. 10): it could be useful a VIMF field to see the prevalence of the flux.

We thank the reviewer for the suggestion. We have now added the VIMF in Fig. 5 and Fig. 10:

**Figure 5:**

[Figure]

**Figure 10:**

[Figure]

Page 10, line 5: the 20% moisture misprice has a similar contribution than those from Central Mediterranean or more. . . Why consider CM and no other sources, for instance the continental ones? Or why the authors do not consider to join CM and WM?

We hypothesize that the Central Mediterranean basin could be in some cases a very important moisture source, especially when a long-distance easterly flow is established. This is quite common, as in the case of October, and anticipating a more relevant role in other events, we explicitly decided to include the Central Mediterranean as an area to consider. In a response to a previous comment we have justified why we have not analyzed other sources, such as the continental ones.

Although we considered both the western and the central Mediterranean as local sources, we found it of great interest to know whether humidity came from the waters closest to the coast

or from a more distant area. Thus, we decided to explicitly separate the Mediterranean according to geographical criteria into a western and a central basin.

Page 11. How the authors could difference from which part of the ST source the final moisture for precipitation comes from? Because the ST contains also moisture that is advected by the AR to northern latitudes. So it seems that the moisture comes only from the eastern part of the Atlantic Ocean of the ST, and over the Sahel region, that it is not completely taken into account (the plot only show a box from 7.5°N. Could be the source of moisture even further south?. That is the problem if the sources were previously predefined.

The only way to know which part of the tropical and subtropical source area moisture originates from would be to subdivide this region into several ones. We could generate source areas as small as the resolution of the model (20 km) allows. However, the more sources there are the higher the computational cost, and as we already argued in another response to a reviewer´s comment, the point of the article is not really to go into such geographic detail, but rather to be able to say whether moisture is from local or remote origin.

The ST source is three-dimensional, so all moisture entering it at any level will be tagged, except for the moisture originating in the other three sources, which are two-dimensional. For this reason, it does not make sense to extend this 3D source further south, since all moisture coming from regions below the south-boundary of the domain, will be tagged upon entering it according to the specified boundary conditions from reanalysis. Therefore, the moisture coming from the part of the Sahel that is not included in the domain, is also being taken into account when advected through the ST source.

Page 11, line20: the omega pattern is typical in a cut-off low formation. The October case appears as a pure cut-off low, but it needs to come from an elongated trough, that develops to a phase of tear-off (when an omega configuration is normal. So the big difference between this November case and the October one is that the COL, in this case, is bigger in size.

In our opinion, there was no COL in the November event, since as shown in the image below (geopotential height and wind at 200 hPa), at the key time of the episode (12 UTC on November 7), the jet stream was not yet completely broken. Therefore, as the upper-level low is not completely uncoupled from the westerlies, we would not call it COL. Perhaps the best term to refer to it would be closed low.

[Figure]

Page 12: the affected areas in relation to the cut-off low position were analyzed in a paper included in the special issue in MAP journal commented previously.

Thank you for the comment. We have already cited that paper in a previous section.

Page 14: after these results I recommend the authors to joint CM and WM.

We disagree. For the reasons stated in responses to previous comments, it seems to us more appropriate to maintain the division.

---

## Author Comment (AC2) · 1 Mar 2019

This is an excellent piece of research that brings a substantial advance to the complex issue of moisture sources related to flood-producing precipitation events over the Mediterranean region. While the writing could be improved, the results are very convincing. I find the methodology particularly praiseworthy. As such, I recommend the article to be accepted after some minor revision.

We would like to thank very much the referee for his kind remarks and positive review. Please, find below the responses to your comments.

General comment: expand the focus.

I agree with the other reviewer that the Authors should consider changing the title. Aside from defining as 'famous' events that may not be known outside the hydrology, engineering, and meteorology communities in Spain, I would rather use the term 'infamous' to describe catastrophic events that have caused death and destruction. Even better, I would avoid 'fame' entirely and perhaps refer to the events as 'catastrophic'. The term 'famous' also appears in page 17, second par.

We agree with the reviewer so the word "famous" is going to be replaced by "catastrophic" in the title. "Famous" will be replaced by "infamous" in the text.

Most important, I suggest to modify the introduction, in order to provide a broader motivation that can make the article relevant to a much larger community. The description of general mechanisms as it is cannot provide objective, generic, absolute 'causes'. Furthermore, the Authors themselves acknowledge that the 2 events are very different one from the other. Therefore, I suggest to broaden the focus of this article, by connecting this work with other research. Conditions for the development of these events surely are strong instability, presence of some circulation that organizes the flow, exploiting orographic contribution. However, the puzzling aspect is that most of these conditions, for example Mediterranean baroclinic cyclones, are often present but are rarely associated with extreme precipitation.

It is only a very small subset of Mediterranean cyclones that cause catastrophic events. Furthermore, in some instance the Mediterranean cyclone could be less relevant than the large-scale southerly flow associated with larger cyclonic circulations outside the Mediterranean. So, the Authors may consider starting with the statement that the presence (or absence) of intense moisture transport anomalies on a very large scale could be the critical, discriminating factor between many situations apparently similar but in which only one produces an extreme precipitation event.

Another suggestion is to think in a more 'global' scale. The Authors' experiments, unlike previous work, are both at very high resolution and encompass a very large domain. As such, they have the possibility of linking Mediterranean floods with the global scale. If this is done, the article will attract a much larger set of readers interested in the subject of tropical-extratropical connections.

As a starting work, consider the final part of Wu et al. (2013). In that work, we linked 2 cases of flood-producing precipitation over Europe with the so-called 6-9 day African Easterly Waves (for the part of the article relevant to your work, please see Section 3d, from page 6765 onwards). The 6-9 waves are different from the well-known 3-6 day waves, because they form to the north of the African Easterly Jet, at the jet level (about 600 hPa) and they travel northward. They can be conceived as a 'relaxation' in the subtropical high pressure that bridges mid latitude low pressure systems with the Inter Tropical Convergence Zone (ITCZ).

From this article's perspective, 6-9 day waves are relevant because they represent a way of connecting tropical moisture generated within the ITCZ with midlatitude systems. If one of such waves acts in phase with a deep midlatitude cyclone, a stream of moisture can leave the ITCZ, travel in a relatively stable area associated with the relaxed subtropical high (and thus without loosing moisture) and 'connect' with the warm advection ahead of a frontal system in the midlatitudes. Then, any mechanism able to concentrate and release this enormous amount of moisture over a small area, can cause a flood-producing precipitation event. Figures 14, 15, and 16 from Wu et al. (2013) illustrate this aspect for extreme precipitation events occurred in 2000 and 2002, respectively. It seems that the plume of moisture associated with Fig.3b, Fig. 5d, in this work, bears remarkable similarities.

Most relevant for this work are also the papers by Knippertz (2003), Knippertz et al. (2003) which connect episodes of extreme precipitation over northwest Africa with anomalous advection from the tropics, and place these into the context of a tropical-extratropical interaction.

Schepanksi and Knippertz (2011) further expand in this direction and finds in the Soudano-Saharan depression a key element connecting tropics with midlatides. We think that all these results are very consistent with each other, and simply focus on different aspects of the moisture transport.

It is important to notice that anomalous moisture advection from the tropical Atlantic has been noted also outside the Mediterranean region: the study of Stohl et al. (2008) identifies a very similar moisture path for precipitation events in Norway, connecting these with the tropical Atlantic.

We fully agree with the general comments provided by the reviewer, and we think it very appropriate to include his recommendations in the text. In order to address these suggestions, we are going to rewrite and expand paragraph one and two of the introduction as follows:

"The Western Mediterranean Region (WMR) is characterized by a high frequency in the occurrence of torrential rainfall episodes and floods that cause severe damages, with a very high social and economic impact (Llasat et al., 2010). The main mechanism generating these heavy precipitation events (HPEs) is the strong instability induced by the warm and moist air that for most of the year sits over the mild Mediterranean waters, along with the presence of a low pressure system (usually produced by a Mediterranean cyclogenesis event) that can trigger convection and organize the flow (Llasat, 2009). Other factors such as the complex orography of the region, often take also a very important role (e.g. Buzzi et al., 1998; Rotunno and Ferretti, 2003). Most cases occur in autumn, when the combination of a still warm sea

surface temperature (after a peak in late summer), and a southward displacement of the jet stream, which usually favours the appearance of Atlantic lows or cut-off-lows (COLs; e.g. Nieto et al., 2005, 2008) affecting the WMR, make this season the most favourable for the development of these extreme events (see Dayan et al., 2015, for a detailed review of the most frequent atmospheric conditions resulting in Mediterranean HPEs).

While factors such as strong instability or the presence of a Mediterranean low in the vicinity are commonly associated with HPEs (Jansa et al., 2001, 2014), the concurrence of these weather features does not ensure the development of extreme precipitation. For example, in autumn, and other seasons too, the presence of Mediterranean cyclones is certainly much more frequent (Campins et al.,2011) than the occurrence of catastrophic flooding episodes (Llasat et al., 2013). Thus, an important question arises: what is the discriminating factor among many apparently similar weather situations where only one produces an HPE? The starting hypothesis of this work is that the factor setting extreme precipitation situations apart is the existence of a very large moisture supply from remote regions outside the Mediterranean. This very humid external influx, when added to local moisture, would yield the enormous amounts of total precipitable water (TPW) needed to produce the rain accumulations commonly recorded in these episodes, which often remind of the values associated with tropical systems. Once sufficient TPW is present, any mechanism able to concentrate and release this moisture over a small area can cause a flood-producing precipitation event. Under this hypothesis, the configuration of the large-scale circulation would therefore be also critical, since it determines whether an intense moisture transport from remote regions can be established or not.

Different studies support the aforementioned idea, especially those that point to the interaction with tropical regions as being key in the development of mid-latitude HPEs. The argument is that when extra-tropical baroclinic low pressure systems descend enough in latitude, they have the chance to capture large amounts of moisture generated in the tropics and advect them into the mid-latitudes and beyond. Thus, a high risk of severe rainfall would be induced if that high moisture content is forced upward by some mechanism, such as orographic lift. The phase of the baroclinic wave should be such allowing this moisture to enter the circulation on the eastern side of the cyclone, often funneled along the cold front, and then be transported poleward resulting in well-known structures such as tropical plumes or atmospheric rivers. Tropical moisture exports have been shown to be important contributors to precipitation in mid-latitude regions in both hemispheres, especially relevant for extreme precipitation (Knippertz and Wernli, 2010; Knippertz et al., 2013). Case studies of HPEs in different parts of the planet, such as the west coast of the United States and Europe, conclude that moisture from tropical and subtropical regions can be essential, comprising most of the TPW feeding these severe episodes (e.g. Stohl et al., 2005; Eiras-Barca et al., 2017). The tropical-extratropical connection in the aforementioned cases occurred through an atmospheric river, advecting highly humid air from lower latitudes to the affected areas. In the WMR, studies for different events also evidence the important role that tropical moisture exports can play in the development of HPEs (e.g. Winschall et al., 2014; Krichak et al., 2015). Some even go further and claim that tropical systems, such as Atlantic hurricanes and their extratropical remnants, can be instrumental, by injecting large amounts of moisture into the Mediterranean Basin (e.g. Pinto et al., 2001; Reale et al., 2001). In the Eastern Mediterranean,

different research have also shown the importance for heavy precipitation of moisture transport from the tropics, sometimes reflected in the formation of tropical plumes (e.g. Ziv, 2001; Rubin et al., 2007). Another example of how tropical-extratropical interactions can trigger Mediterranean severe rains come from Wu et al. (2013), who found that the interaction of the (6-9 days) African easterly wave with mid-latitude low pressure systems resulted in large moisture exports from the tropics that were fundamental in producing the precipitation inducing floods in 2000 and 2002. All these studies reinforce the need to look at the problem of extreme precipitation in the Mediterranean region from a more global perspective and discard a local or regional view, particularly in the case of moisture origin.

However, in the ample literature analyzing the different contributors to the genesis of HPEs in the Western Mediterranean, moisture as a key factor is sometimes undervalued or not considered in depth, often assuming that the high values of TPW involved in these events originate locally at low levels from sea evaporation. But, where does such large amount of water vapour really come from? Is it evaporation in the Mediterranean the main source or, on the contrary, does most of the moisture in precipitation originate remotely?"

Nieto, R., Gimeno, L., de La Torre, L., Ribera, P., Gallego, D., García-Herrera, R., ... & Lorente, J. (2005). Climatological features of cutoff low systems in the Northern Hemisphere. Journal of Climate, 18(16), 3085-3103.

Nieto, R., Sprenger, M., Wernli, H., Trigo, R. M., & Gimeno, L. (2008). Identification and Climatology of Cut-off Lows near the Tropopause. Annals of the New York Academy of Sciences, 1146(1), 256-290.

Dayan, U., Nissen, K., & Ulbrich, U. (2015). Atmospheric conditions inducing extreme precipitation over the eastern and western Mediterranean. Natural Hazards & Earth System Sciences Discussions, 3(11).

Jansa, A., Genoves, A., Picornell, M. A., Campins, J., Riosalido, R., & Carretero, O. (2001). Western Mediterranean cyclones and heavy rain. Part 2: Statistical approach. Meteorological Applications, 8(1), 43-56.

Jansa, A., Alpert, P., Arbogast, P., Buzzi, A., Ivancan-Picek, B., Kotroni, V., ... & Speranza, A. (2014). MEDEX: a general overview. Natural Hazards and Earth System Sciences, 14(8), 1965-1984.

Campins, J., Genovés, A., Picornell, M. A., & Jansà, A. (2011). Climatology of Mediterranean cyclones using the ERA-40 dataset. International Journal of Climatology, 31(11), 1596-1614.

Llasat, M. C., Llasat-Botija, M., Petrucci, O., Pasqua, A. A., Rosselló, J., Vinet, F., & Boissier, L. (2013). Towards a database on societal impact of Mediterranean floods within the framework of the HYMEX project. Natural Hazards and Earth System Sciences, 13(5), 1337-1350.

Knippertz, P., & Wernli, H. (2010). A Lagrangian climatology of tropical moisture exports to the Northern Hemispheric extratropics. Journal of Climate, 23(4), 987-1003.

Knippertz, P., Wernli, H., & Gläser, G. (2013). A global climatology of tropical moisture exports. Journal of Climate, 26(10), 3031-3045.

Stohl, A., Forster, C., & Sodemann, H. (2008). Remote sources of water vapor forming precipitation on the Norwegian west coast at 60 N–a tale of hurricanes and an atmospheric river. Journal of Geophysical Research: Atmospheres, 113(D5).

Eiras-Barca, J., Dominguez, F., Hu, H., Garaboa-Paz, D., & Miguez-Macho, G. (2017). Evaluation of the moisture sources in two extreme landfalling atmospheric river events using an Eulerian WRF tracers tool. Earth System Dynamics, 8(4), 1247.

Winschall, A., Sodemann, H., Pfahl, S., & Wernli, H. (2014). How important is intensified evaporation for Mediterranean precipitation extremes?. Journal of Geophysical Research: Atmospheres, 119(9), 5240-5256.

Krichak, S. O., Barkan, J., Breitgand, J. S., Gualdi, S., & Feldstein, S. B. (2015). The role of the export of tropical moisture into midlatitudes for extreme precipitation events in the Mediterranean region. Theoretical and applied climatology, 121(3-4), 499-515.

Pinto, J. G., Klawa, M., Ulbrich, U., Rudari, R., & Speth, P. (2001, October). Extreme precipitation events over northwest Italy and their relationship with tropical–extratropical interactions over the Atlantic. In Proceedings of the third EGS Plinius Conf. on Mediterranean Storms, Baja Sardinia, Italy, GNDCI Publication (No. 2560, pp. 321-332).

Reale, O., Feudale, L., & Turato, B. (2001). Evaporative moisture sources during a sequence of floods in the Mediterranean region. Geophysical research letters, 28(10), 2085-2088.

Ziv, B. (2001). A subtropical rainstorm associated with a tropical plume overAfrica and the Middle-East. Theoretical and Applied Climatology, 69(1-2), 91-102.

Rubin, S., Ziv, B., & Paldor, N. (2007). Tropical plumes over eastern North Africa as a source of rain in the Middle East. Monthly Weather Review, 135(12), 4135-4148.

Wu, M. L. C., Reale, O., & Schubert, S. D. (2013). A characterization of African easterly waves on 2.5–6-day and 6–9-day time scales. Journal of Climate, 26(18), 6750-6774.

Minor comments:

The WVT could become a formidable tool, particularly because is coupled with the WRF, which is very well known and used worldwide. The Authors should consider distributing it, either through their own portal, or in collaboration with an WRF development team. It would gather widespread attention if it became an easily accessible methodology. I find particularly important, compared to earlier studies, the ability of investigating sources on a 3D scale.

Thank you for your suggestion. As we have also commented to the other reviewer, we are currently considering making the code open access.

Figures. The clarity of the figures illustrating the synoptic situation could be improved. Even coastlines and geographic features are barely detectable. In Figure 3a, I suggest blanking out the temperatures around 0C (such as -2 2) so as to have a white strip between  cold  and

warmer air, and use less intense colors. If done in GrADS, the Authors could consider the use of transparencies which is now an option and allows more readable plots. Otherwise, just use lighter colors.

In figure 3b, I suggest to blank out completely the values of tpw less than 10mm. These are not relevant to this work, indicate simply drier air, and by eliminating them the emphasis would be given to the huge moisture plume stretching from the subtropical Atlantic towards the Iberian peninusula.

Similar suggestions for Figure 8.

We have modified Figure 3 and Figure 8 following the reviewer's suggestions:

**Figure 3**

[Figure]

**Figure 8**

[Figure]

In the text, there are some sentences whose clarity could be improved. See for example page 4, lines 7-8, d

The sentence will be clarified.

Page 18, line 14.. a semantic issue.. Instead of 'verify the hypothesis' .. . Consider something like 'corroborate the idea that remote sources of moisture from the tropics contribute to an important fraction of extreme precipitaiton events in the midlatitudes..'

The sentence will be corrected as suggested.

In summary, aside from these relatively minor suggestions, I believe that the article is a great contribution to precipitation research, and I recommend acceptance after minor revision. However, I believe that the article would benefit by placing the results into the broader context of tropical-extratropical interactions and large scale advection of tropical moisture.

Thank you again for your positive comments.

---

## Author Comment (AC3) · 9 Mar 2019

In the previous comment, we proposed to modify and extend the first part of the introduction following the reviewer's suggestions. After those changes, the third paragraph of the introduction dealt with the importance of tropical-extratropical interactions for mid-latitudes extreme precipitation:

"Different studies support the aforementioned idea, especially those that point to the interaction with tropical regions as being key in the development of mid-latitude HPEs. The argument is that when extra-tropical baroclinic low pressure systems descend enough in latitude, they have the chance to capture large amounts of moisture generated in the tropics and advect them into the mid-latitudes and beyond. Thus, a high risk of severe rainfall would be induced if that high moisture content is forced upward by some mechanism, such as orographic lift. The phase of the baroclinic wave should be such allowing this moisture to enter the circulation on the eastern side of the cyclone, often funneled along the cold front, and then be transported poleward resulting in well-known structures such as tropical plumes or atmospheric rivers. Tropical moisture exports have been shown to be important contributors to precipitation in mid-latitude regions in both hemispheres, especially relevant for extreme precipitation (Knippertz and Wernli, 2010; Knippertz et al., 2013). Case studies of HPEs in different parts of the planet, such as the west coast of the United States and Europe, conclude that moisture from tropical and subtropical regions can be essential, comprising most of the TPW feeding these severe episodes (e.g. Stohl et al., 2005; Eiras-Barca et al., 2017). The tropical-extratropical connection in the aforementioned cases occurred through an atmospheric river, advecting highly humid air from lower latitudes to the affected areas. In the WMR, studies for different events also evidence the important role that tropical moisture exports can play in the development of HPEs (e.g. Winschall et al., 2014; Krichak et al., 2015). Some even go further and claim that tropical systems, such as Atlantic hurricanes and their extratropical remnants, can be instrumental, by injecting large amounts of moisture into the Mediterranean Basin (e.g. Pinto et al., 2001; Reale et al., 2001). In the Eastern Mediterranean, different research have also shown the importance for heavy precipitation of moisture transport from the tropics, sometimes reflected in the formation of tropical plumes (e.g. Ziv, 2001; Rubin et al., 2007). Another example of how tropical-extratropical interactions can trigger Mediterranean severe rains come from Wu et al. (2013), who found that the interaction of the (6-9 days) African easterly wave with mid-latitude low pressure systems resulted in large moisture exports from the tropics that were fundamental in producing the precipitation inducing floods in 2000 and 2002. All these studies reinforce the need to look at the problem of extreme precipitation in the Mediterranean region from a more global perspective and discard a local or regional view, particularly in the case of moisture origin."

We think it is too early to talk about this now and we prefer to postpone this topic for another article we are working on. In the later article, which we are about to finish, we deal with a much larger number of cases (all major flooding events in the last 35 years), so we hope to draw much more general conclusions. It is at that point that we will attempt to link Mediterranean heavy precipitation events with tropical-extratropical connections, and we will

compare the results obtained with those obtained by other authors. In short, we prefer to leave this topic for the closing of our project (my PhD thesis) rather than for the opening. For this reason, we now propose deleting the aforementioned third paragraph.

With this last correction and some other minor modification, the first part of the introduction would read as follows:

[revised manuscript text omitted]